

SciPost Phys. Comm. Rep. 4 (2024)

# LHC EFT WG note: SMEFT predictions, event reweighting, and simulation

Alberto Belvedere[1], Saptaparna Bhattacharya[2], Giacomo Boldrini[3], Suman Chatterjee[4], Alessandro Calandri[5], Sergio Sánchez Cruz[6], Jennet Dickinson[7], Franz J. Glessgen[5], Reza Goldouzian[8], Alexander Grohsjean[1], Laurids Jeppe[1], Charlotte Knight[9], Olivier Mattelaer[10], Kelci Mohrman[11], Hannah Nelson[8], Vasilije Perovic[5], Matteo Presilla[12], Robert Schoefbeck[4] and Nick Smith[7]

**1** Deutsches Elektronen-Synchrotron (DESY), Hamburg, Germany
**2** Northwestern University, Evanston, Illinois, and
Wayne State University, Detroit, Michigan, United States
**3** Laboratoire Leprince-Ringuet (LLR), Ècole Polytechnique, Palaiseau Cedex, France
**4** Institute for High Energy Physics of the Austrian Academy of Sciences, Vienna, Austria
**5** ETH Zürich, Zürich, Switzerland
**6** CERN, Geneva, Switzerland
**7** Fermi National Accelerator Laboratory (FNAL), Batavia, Illinois, United States
**8** University of Notre Dame, Notre Dame, Indiana, United States
**9** Imperial College, London, United Kingdom
**10** Catholic University of Louvain, Louvain, Belgium
**11** University of Florida, Gainesville , United States
**12** Institute for Experimental Particle Physics (ETP),
Karlsruhe Institute of Technology (KIT), Karlsruhe, Germany

## Abstract

This note provides a comprehensive overview of tools for predicting observables in the Standard Model effective field theory (SMEFT) at both tree level and one loop using event generators. We evaluate three primary methodologies–event reweighting, separate simulation of squared matrix elements, and full SMEFT process simulation–focusing on their statistical performance, computational efficiency, and potential biases. Each approach is assessed in terms of its accuracy, highlighting trade-offs between precision and resource demands. Practical insights into their applicability for high-energy physics analyses are offered, with particular attention to processes where SMEFT effects are significant. Additionally, we discuss the role of helicity in reweighting strategies and its impact on the quality of predictions. By comparing the methods across various LHC processes, this note provides guidance for selecting the most effective strategy for various SMEFT studies, ensuring robust predictions while optimizing computational resources.

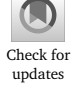

# 1   Introduction and motivation

The Standard Model effective field theory (SMEFT) [1–5] provides a low-energy parametrization of phenomena beyond the Standard Model (SM) in terms of Wilson coefficients (WC). The WCs are the prefactors of symmetry-preserving local field operators in the SMEFT Lagrangian, whose measurement allows for discriminating between different UV models.

The main organizing principle of the SMEFT operators is their mass dimension, starting at six for phenomena relevant at the LHC. Accurate predictions for high-dimensional SMEFT analyses require a versatile and robust toolkit, whose ranges of applicability and potential shortfalls must be understood in detail. Earlier notes of the LHC EFT working group (WG) cover several important steps forward in this regard. The relation between hypothetical high-scale physics and the SMEFT operators can be obtained by matching the integrated effect of the high-scale beyond the Standard Model (BSM) phenomena to the SMEFT WCs. Automated tools for this matching are reviewed in Ref. [6]. A review of experimental SMEFT measurements and observables is provided in Ref. [7]. Finally, strategies for treating uncertainties related to the truncation of the effective field theory (EFT) expansion at finite mass dimension are discussed in Ref. [8]. In this work, we do not quote the range of validity of the EFT expansion for the distributions used in the comparisons, because the consistency of the computational strategies is unaffected by the validity of the expansion.

This note serves as a guide to obtaining SMEFT predictions from event generators for usage in LHC data analyses. It assesses the quality of reweighting- and sampling-based strategies for obtaining generator-level predictions by comparing them to a reference strategy of "direct" simulation at a specific fixed parameter point. It also aims to highlight best practices and document common pitfalls but does not establish authoritative guidelines.

Section 2 discusses the different methodologies for obtaining simulated SMEFT predictions in terms of the WCs. In Sec. 3, the role of the initial- and final-state helicities is clarified. Best practices and common pitfalls are summarized in Sec. 4. The main body of the work, a comparison of SMEFT predictions obtained from different methods, is presented in Sec. 5. Section 6 gives a summary.

## 2 Strategies for simulated predictions

Our starting point is the SMEFT Lagrangian, which extends the SM by introducing $M(d)$ symmetry-preserving operators $\mathcal{O}$ with mass dimension $d > 4$,

$$\mathcal{L}_{\text{SM-EFT}} = \mathcal{L}_{\text{SM}}^{(4)} + \sum_{d>4} \sum_{a=1}^{M(d)} \frac{\theta_a \mathcal{O}_{(d)}^a}{\Lambda^{d-4}}, \tag{1}$$

where $\theta_a$ represents the WCs. Equation 1 captures non-resonant phenomena beyond the SM (BSM) at energy scales below an unknown new-physics threshold. In practice, a normalization scale $\Lambda$ is introduced, typically fixed at 1 TeV. Since a generic SMEFT differential cross-section with single-operator insertions can be written as

$$d\sigma(\boldsymbol{\theta}) \propto \left| \mathcal{M}_{\text{BSM}}(\boldsymbol{z}_p) \right|^2 d\boldsymbol{z}_p = \left| \mathcal{M}_{\text{SM}}(\boldsymbol{z}_p) + \frac{1}{\Lambda^2} \sum_{a=1}^{M} \theta_a \mathcal{M}_{\text{EFT}}^a(\boldsymbol{z}_p) \right|^2 d\boldsymbol{z}_p, \tag{2}$$

the SMEFT predictions for event rates at the parton level, with momenta $\boldsymbol{z}_p$, can be expressed as polynomials in the WCs. The matrix elements (MEs) for the SM and SMEFT are denoted by $\mathcal{M}_{\text{SM}}$ and $\mathcal{M}_{\text{EFT}}$, respectively.

In Eq. 2 and in the following, we collectively label observable features by $\boldsymbol{x}$ and unobservable (latent) variables by $\boldsymbol{z}$. The only exception is the Bjorken scaling variables, where we maintain the convention and denote them as $x_{\text{Bjorken},1}$ and $x_{\text{Bjorken},2}$, although these are part of $\boldsymbol{z}$. At the parton level, $\boldsymbol{z}_p$ includes the four-momenta of the external partons and, generically, the helicity configuration denoted by $h$.

Whether or not $h$ is considered part of $\boldsymbol{z}_p$ is a matter of choice, with important practical implications for the reweighting-based strategies discussed in Sec. 3. In the former case, we have

$$d\boldsymbol{z}_p = f_1(x_{\text{Bjorken},1}, \mu_F) f_2(x_{\text{Bjorken},2}, \mu_F) d\Omega_{\text{PS}}^{(h)}, \tag{3}$$

where $f_i(x_{\text{Bjorken},i}, \mu_F)$ represents the parton distribution function (PDF) for a factorization scale $\mu_F$. The per-helicity kinematic phase space element of the external particles is denoted by $d\Omega_{\text{PS}}^{(h)}$ and includes the measure over the Bjorken variables, such that

$$d\sigma(\boldsymbol{\theta}) \propto \left| \mathcal{M}_{\text{BSM}}(\boldsymbol{z}_p, h) \right|^2 f_1(x_{\text{Bjorken},1}, \mu_F) f_2(x_{\text{Bjorken},2}, \mu_F) d\Omega_{\text{PS}}^{(h)}. \tag{4}$$

If helicity information is not available or dropped, it is excluded from the parton-level phase-space definition. For instance, when the ME generator does not include helicity information, the helicity dependence of the $|\mathcal{M}|^2$ terms is summed, and we have

$$d\sigma(\boldsymbol{\theta}) \propto \left( \sum_h \left| \mathcal{M}_{\text{BSM}}(\boldsymbol{z}_p, h) \right|^2 f_1(x_{\text{Bjorken},1}, \mu_F) f_2(x_{\text{Bjorken},2}, \mu_F) \right) d\Omega_{\text{PS}}, \tag{5}$$

with the important distinction that $d\Omega_{\text{PS}}$ now multiplies a sum over $h$. Either way, automated ME generators produce a numerical code for Eq. 2, which can be efficiently re-evaluated for different $\boldsymbol{\theta}$ for a given $\boldsymbol{z}_p$. This computational efficiency forms the basis for the reweighting strategies discussed in this note.

In this note, we quantitatively compare three different strategies for obtaining SMEFT predictions via event simulation using the SMEFT Lagrangian in Eq. 1. All studies truncate the perturbative expansion at leading order (LO) or next-to-LO (NLO) in QCD. The simplest procedure chooses the desired value of $\boldsymbol{\theta}$ and samples the SMEFT model at this parameter point ("direct simulation"). While this approach is conceptually straightforward and serves

as our reference, it is not computationally efficient for most practical applications, as constraints on WCs require comparing likelihoods for arbitrary $\boldsymbol{\theta}$, typically exceeding available computational resources for event simulation.

There are two main strategies for obtaining parametrized predictions. Firstly, the SMEFT ME-squared terms in Eq. 2 can be expanded, and the terms corresponding to the same polynomial coefficient in $\boldsymbol{\theta}$ can be sampled separately and independently ("separate simulation"). Events from the resulting samples can then be weighted according to the desired value of $\boldsymbol{\theta}$. If we denote the event sample at the SM by $S_0$, and the event samples obtained from the linear terms in Eq. 2 by $S_a$, a yield $\lambda_{\Delta z}$ in a small phase space volume $\Delta z$ around the parton-level configuration $z_p$ is predicted to be

$$\lambda_{\Delta z}(\boldsymbol{\theta}) = \sum_{z_i \in \Delta z \cap S_0} w_{i,0} + \sum_{a=1}^{M} \theta_a \sum_{z_i \in \Delta z \cap S_a} w_{i,a} + \sum_{\substack{a,b=1 \\ a \geq b}}^{M} \theta_a \theta_b \sum_{z_i \in \Delta z \cap S_{ab}} w_{i,ab} \,, \tag{6}$$

where the constant weights $w_{i,0}$, $w_{i,a}$, and $w_{i,ab}$ are obtained from the generator. The normalization can be chosen as

$$\mathcal{L}\sigma(\boldsymbol{\theta}) = \sum_{i \in S_0} w_{i,0} + \sum_{a=1}^{M} \theta_a \sum_{i \in S_a} w_{i,a} + \sum_{\substack{a,b=1 \\ a \geq b}}^{M} \theta_a \theta_b \sum_{i \in S_{ab}} w_{i,ab} \,, \tag{7}$$

where $\mathcal{L}$ is the integrated luminosity and $\sigma(\boldsymbol{\theta})$ represents the inclusive cross-section.

Secondly, the per-event parton-level configuration of an event from a sample obtained with a specific SMEFT parameter reference point $\boldsymbol{\theta}_0$, not necessarily the same as the SM at $\boldsymbol{\theta}_0 = \mathbf{0}$, can be used to re-evaluate Eq. 2 at different values of $\boldsymbol{\theta}$. Since the differential cross section is a quadratic function of the WCs, a small number of evaluations can be used to determine a polynomial that parametrizes the weight of the event when computing the predicted yield as

$$\lambda_{\Delta z}(\boldsymbol{\theta}) = \sum_{z_i \in \Delta z} w_i(\boldsymbol{\theta}) = \sum_{z_i \in \Delta z} \left( w_{i,0} + \sum_{a=1}^{M} \theta_a w_{i,a} + \sum_{\substack{a,b=1 \\ a \geq b}}^{M} \theta_a \theta_b w_{i,ab} \right). \tag{8}$$

To determine the per-event polynomial coefficients $w_{i,0}$, $w_{i,a}$, and $w_{i,ab}$ from the event generator, a set of $k = 1, \ldots, K$ different SMEFT base points $\boldsymbol{\theta}^{(k)}$ is needed, and $K$ must be at least equal to the number of degrees of freedom, that is, $N = 1 + M + \frac{1}{2}M(M+1)$ at quadratic order. If we let an index $n$ enumerate the constant term, the $M$ linear terms, and the $\frac{1}{2}M(M+1)$ quadratic terms, we can take the constant factors from Eq. 8 to form the $K \times N$ matrix $\Theta_n^{(k)} = \{1, \theta_a^{(k)}, \theta_a^{(k)}\theta_b^{(k)}\}$.

For $K = N$, i.e., if we have obtained just enough coefficients $w_i(\boldsymbol{\theta}^{(k)})$ at the base points $\boldsymbol{\theta}^{(k)}$, we can uniquely solve the linear set of equations,

$$w_i(\boldsymbol{\theta}^{(k)}) = \sum_n \Theta_n^{(k)} w_{i,n} \,, \tag{9}$$

for the polynomial coefficients $w_{i,n}$ of Eq. 8 in terms of the event weights provided by the generator. Again, the index $n$ labels the constant term, the $M$ linear terms, and the quadratic terms. For $K \geq N$, the polynomial coefficients can be determined if the $K \times N$ matrix $\Theta_n^{(k)}$ has full rank.

In the case of reweighting, it is an important practical distinction whether the generator computes the ME-squared separately for each helicity configuration (helicity-aware, Eq. 4) or whether it first sums over helicities (helicity-ignorant, Eq. 5). In the former case, the simulated helicity contributions are accurately predicted at each stage. In the latter case, only the sum of the SMEFT predictions over all helicity configurations is correct. The advantage of helicity-ignorant reweighting is that it avoids large weights when a SMEFT operator introduces helicity configurations that are suppressed in the SM. In both cases, the normalization of the reweighted samples can be written as

$$\mathcal{L}\sigma(\boldsymbol{\theta}) = \sum_{i \in S} w_i(\boldsymbol{\theta}). \tag{10}$$

There are also important differences between the "reweighted simulation" in Eq. 8 and the separate simulation in Eq. 6. Firstly, there is no stochastic independence in the constant, linear, and quadratic terms when reweighting. For each event, the probabilistic mass of its concrete parton-level configuration–i.e., the event's weight when computing yields–is known across all SMEFT parameter space, with potential benefits for machine-learning applications [9–12]. In contrast, the separate simulation of the different ME-squared terms predicts the constant, linear, and quadratic terms with uncorrelated statistical uncertainties, potentially increasing the CPU demand for a given requirement on statistical precision. Secondly, the independent sampling does not depend on a reference point. In practice, event reweighting can lead to large weights in regions of phase space where the parton-level differential cross sections differ significantly between $\boldsymbol{\theta}$ and the reference $\boldsymbol{\theta}_0$. This effect can be particularly severe when SMEFT operators introduce, for example, helicity configurations that are not present in the SM and helicity-aware reweighting is used.

Finally, let us clarify the relation to parametrized SMEFT predictions at lower-level representations of the simulated data, e.g., after detector simulation. Does it have implications for reweighting? Following the ME generators providing the parton-level differential cross sections, a hierarchical sequence of staged computer codes is used to simulate phenomena at lower energy scales and using, typically, much higher-dimensional representations of the events. These stages include the parton shower with ME-matching and merging procedures, the hadronization of the shower algorithm's output, the detector interactions, and the event reconstruction. Many of these stages are at least partially stochastic.

Provided $S$ is sufficiently large for the statistical uncertainty in $\lambda_{\Delta z}$ to be acceptably small for any $\Delta z$ in the phase space covered by $S$, we use Eq. 6 or Eq. 8 to approximate the parton-level differential cross section as

$$\frac{1}{\mathcal{L}\sigma(\boldsymbol{\theta})} \frac{\lambda_{\Delta z}(\boldsymbol{\theta})}{\Delta z} \approx \frac{1}{\sigma(\boldsymbol{\theta})} \frac{\mathrm{d}\sigma(z_p|\boldsymbol{\theta})}{\mathrm{d}z_p} = p(z_p|\boldsymbol{\theta}), \tag{11}$$

where the l.h.s. and r.h.s. of the first equation each represent a ratio of quadratic polynomials. The last equality interprets the normalized differential cross section as the parton-level probability density function. As illustrative examples of how to transition to the detector level, we first define the particle-level $z_{\mathrm{ptl}}$, comprising stable generated particles after hadronization and before interaction with the detector material. Secondly, the detector-level representation $x_{\mathrm{det}}$ of the simulated processes shall consist of, for example, jets, b-tagged jets, leptons, and other reconstructed high-level objects. This representation is the simulated equivalent of the detector-level observation of real data in a generic analysis. Eq. 11 can then be used to express, e.g., the detector-level cross section as

$$\frac{\mathrm{d}\sigma(x|\boldsymbol{\theta})}{\mathrm{d}x} = \int \mathrm{d}z_{\mathrm{ptl}} \int \mathrm{d}z_p \, p(x|z_{\mathrm{ptl}}) p(z_{\mathrm{ptl}}|z_p) \frac{\mathrm{d}\sigma(z_p|\boldsymbol{\theta})}{\mathrm{d}z_p}. \tag{12}$$

The conditional distribution $p(z_\text{ptl}|z_p)$ is sampled by the shower simulation, the hadronization model, and the matching and merging procedures. The conditional distribution $p(x|z_\text{ptl})$ governs the detector simulation and the event reconstruction. Both distributions are intractable, meaning they can be sampled for a fixed conditional configuration but cannot be evaluated as a function of the condition for a fixed sampling instance. Nevertheless, it has been shown that intractable factors cancel, provided that the WCs do not modify these distributions [9, 13–16].

Concretely, the probability to obtain a certain observation $x$ given a particle-level configuration $z_\text{ptl}$ shall not depend on the WCs, and neither shall the probability to observe a certain particle-level configuration when a parton-level event is given. In this case, dividing both sides by the total cross section and using Eq. 11, we can trivially re-express the detector-level probability density in terms of the parton-level one. The conditional sequence relating the parton level with the detector level through the particle level could be more refined, with more intermediate integrations in Eq. 12, but as long as the SMEFT effects do not affect anything other than $p(z_p|\theta)$, it follows that we can approximate any detector-level yield $\lambda_{\Delta x}$ from separate simulation as

$$\lambda_{\Delta x}(\theta) = \sum_{x_i \in \Delta x \cap S_0} w_{i,0} + \sum_{a=1}^{M} \theta_a \sum_{x_i \in \Delta x \cap S_a} w_{i,a} + \sum_{\substack{a,b=1 \\ a \geq b}}^{M} \theta_a \theta_b \sum_{x_i \in \Delta x \cap S_{ab}} w_{i,ab}, \quad (13)$$

using the *same* per-event weights as in Eq. 6. The corresponding prediction for the case of event reweighting is

$$\lambda_{\Delta x}(\theta) = \sum_{x_i \in \Delta x} \left( w_{i,0} + \sum_{a=1}^{M} \theta_a w_{i,a} + \sum_{\substack{a,b=1 \\ a \geq b}}^{M} \theta_a \theta_b w_{i,ab} \right), \quad (14)$$

again using the same weight functions as in Eq. 8.

To the extent that the intractable conditional likelihoods do not depend on the WCs, we can ignore the level at which we obtain the predictions and simply accumulate the event weight polynomials in bins defined by $x$. In the case of event reweighting, we can furthermore interpret the $w_i(\theta)$ as the total cross section multiplied by the per-event likelihood of the joint observed and generated features,

$$w_i(\theta) = \sigma(\theta) p(x_i, z_{\text{ptl},i}, z_{p,i}|\theta), \quad (15)$$

which agrees with the joint likelihood in Ref. [9] up to an overall cross-section normalization. The conceptual simplification of the reweighting strategy then appears in the ratio

$$\frac{w_i(\theta)}{w_i(\text{SM})} = \frac{\sigma(\theta)}{\sigma(\text{SM})} \frac{p(x_i|z_{\text{ptl},i})p(z_{\text{ptl},i}|z_{p,i})}{p(x_i|z_{\text{ptl},i})p(z_{\text{ptl},i}|z_{p,i})} \frac{p(z_{p,i}|\theta)}{p(z_{p,i}|\text{SM})} = \frac{|\mathcal{M}(\theta)|^2(z_{p,i})}{|\mathcal{M}(\text{SM})|^2(z_{p,i})}, \quad (16)$$

via the cancellation of the extremely complicated, usually intractable, conditional likelihood factors $p(x_i|z_{\text{ptl},i})$ and $p(z_{\text{ptl},i}|z_{p,i})$. Once $w_i(\text{SM})$ are known for an event sample, the easily calculable ME-squared ratios are enough to obtain detector-level predictions for any values of the WCs.

## 3 Helicity aware and helicity ignorant reweighting

Any reweighting method consists of modifying the weight of a parton-level event such that the resulting weighted event sample reproduces an alternative scenario, leveraging the statistical

power of a given event sample, possibly removing the need for a dedicated shower- and detector simulation. At LO, ME generators customarily include the helicity configuration associated with the events, even when using Eq. 5. For the nominal simulation, for example, at the SM parameter point, the MADGRAPH5_AMC@(N)LO v2.6.5 event generator [17] selects helicity configurations randomly according to the probability

$$p(h|\boldsymbol{z}_p, \text{nom}) = \frac{\left|\mathcal{M}_{\text{nom}}(\boldsymbol{z}_p, h)\right|^2}{\sum_h \left|\mathcal{M}_{\text{nom}}(\boldsymbol{z}_p, h)\right|^2}, \tag{17}$$

where $\left|\mathcal{M}_{\text{nom}}(\boldsymbol{z}_p, h)\right|^2$ is the squared amplitude for a given helicity configuration $h$, comprising all initial- and final-state particles. Helicity-aware reweighting at LO to an alternative parameter point is implemented by modifying the event weights by a factor

$$w_{\text{alt}} = w_{\text{nom}} \frac{\left|\mathcal{M}_{\text{alt}}(\boldsymbol{z}_p, h)\right|^2}{\left|\mathcal{M}_{\text{nom}}(\boldsymbol{z}_p, h)\right|^2}, \tag{18}$$

while helicity-ignorant reweighting amounts to

$$w_{\text{alt}} = w_{\text{nom}} \frac{\sum_h \left|\mathcal{M}_{\text{alt}}(\boldsymbol{z}_p, h)\right|^2}{\sum_h \left|\mathcal{M}_{\text{nom}}(\boldsymbol{z}_p, h)\right|^2}. \tag{19}$$

A few remarks are in order regarding the range of validity of these methods. Firstly, even if the method is correct in the asymptotic limit of infinite sample size, a real-world application is limited by the size of $p(h|\boldsymbol{z}_p, \text{alt})/p(h|\boldsymbol{z}_p, \text{nom})$ as a function of the parton-level momenta $\boldsymbol{z}_p$. If the alternate scenario strongly differs in terms of helicity configurations or kinematic dependence, this ratio can become very large. Since the statistical power of the helicity-aware reweighted sample corresponds to the nominal sample, the relative statistical uncertainty in the affected phase-space can grow arbitrarily, sometimes entirely removing the feasibility of helicity-aware reweighting. In practice, this is reflected by large event weights.

Secondly, we note that the requirement of a similar phase-space density of the alternate and the nominal hypothesis applies beyond SMEFT reweighting. For example, reweighting cannot be used for scanning mass values far outside of the width of a resonance.

Finally, we remark that, similar to the case of helicity, a choice is needed for reweighting according to the (leading) color assignment of an event, which must be defined in the presence of a mixed perturbative expansion (*e.g.*, the VBF process at LO with both QCD and QED amplitudes included). So far, only color-ignorant reweighting has been implemented, which, in fact, limits the applicability of reweighting to BSM models that leave the relative contributions of color assignments within a process unaffected by the WCs. Color-aware reweighting is a theoretical possibility–though currently not implemented–with the same limitations and advantages as in the case of helicity, as far as the hard-scatter parton level is concerned. The color assignment, however, has a substantial and direct impact on the parton-shower simulation, warranting careful and process-dependent validation in cases where, e.g., four-fermion operators are used.

The particular case of mixed expansion not only creates an issue for the color assignment at LO, but also in handling the reweighting at NLO accuracy. For technical reasons, in the presence of a mixed expansion, each ME is separated into terms with the same power of the coupling constants. A correct reweighting procedure then requires that each order is reweighted by the corresponding ME. Currently, this is not implemented in the MADGRAPH5_AMC@NLO reweighting tool.

# 4 Best practices

We next address common challenges and pitfalls encountered during the studies presented in Sec. 5. It should, therefore, be understood that the list is not exhaustive.

**Renormalization and factorization scales choice.** It is customary to employ a dynamic scale for various generated samples, often opting for the CKKW-L clustering algorithm's scale choice [18], where only clustering compatible with the current integration channel is permitted. The event-specific nature of this scale choice, dependent on both the channel of integration and the event generation method [19], poses a potential issue for consistency tests comparing direct simulation, separate simulation, and reweighting. While this is not an issue for the validity of the prediction, closure tests may only be consistent up to scale variations when employing CKKW-L clustering. Conversely, fixed scale choices determined solely by the event's kinematics remain unaffected. Unless noted otherwise, the scale choices in the closure tests in Sec. 5, corresponding to $H_T/2$ by default, avoid this problem.

**On the NLO SMEFT simulation.** Another important aspect to consider concerns the perturbative order of the MC simulation. While an NLO QCD calculation generally represents the best solution, NLO QCD SMEFT simulations with MADGRAPH5_AMC@NLO can involve challenges with SMEFT operators involving electroweak vertices. These complications stem from the fact that, at the time of writing, MADGRAPH5_AMC@NLO SMEFT calculations can account for NLO QCD effects but cannot account for QED loops. For this reason, restricting the QED coupling order of NLO QCD samples is required.

This restriction does not permit tree-level diagrams with a QED order greater than or equal to two plus the lowest QED order tree-level diagrams to enter; such tree-level contributions are not permitted because they would enter with the same QED order as a QED loop added to the lowest QED order diagrams. SMEFT couplings are assigned QED orders (which are somewhat arbitrary and can differ between UFO models [20]), so tree-level diagrams (with QED order larger than the QED order cutoff imposed for NLO QCD calculations) must be excluded in NLO QCD calculations. However, LO calculations do not require restrictions on the QED order, so the matched LO approach, where LO samples with extra QCD emissions are taken as a proxy for partial NLO QCD effects (see below), does not involve this limitation.

When generating NLO SMEFT samples with operators and processes involving electroweak vertices, it is advisable to also generate LO samples (without QED order restrictions) to ensure that any SMEFT effects excluded at NLO by the QED order restriction are fully understood.

**SMEFT simulation with extra partons.** When NLO samples are not available, it can be beneficial to include an additional parton in the LO ME calculation. Not only does this help to provide more accurate modeling of SM kinematics, but it can also impact the SMEFT dependence of processes on certain Wilson coefficients [20], though careful validation is important. These effects are primarily due to the additional initial states that become available with the inclusion of an additional parton, but other factors (such as the topology of diagrams, energy scaling of vertices, and interference effects) can also be relevant.

Since it is difficult to predict which combinations of processes and operators will be strongly impacted by the inclusion of the additional parton, it is beneficial to include the extra parton whenever possible to avoid inadvertently neglecting relevant SMEFT contributions. With the matched LO approach, the harder emission is handled with MADGRAPH5_AMC@LO (with the extra parton explicitly included in the ME calculation) while softer emission is handled by the parton shower; a matching procedure (e.g. the $k_T$-jet version of the MLM matching scheme [21]) is required to remove the overlap between the two regions. This matching procedure involves choosing cutoff scales for the ME and parton shower, and it is important to validate that the choices for the values of these cutoff scales allow the ME generator and parton shower simulation to smoothly fill the overlapping phase space.

To perform such validation, it is useful to study differential jet rate (DJR) distributions [22, 23]; smooth transitions between the $n$ and $n+1$ curves in a DJR distribution is an indication that the chosen matching scales have allowed the ME generator and parton shower simulation to work together smoothly in the overlapping space.

Performing matching with SMEFT samples can introduce an additional complication. Since SMEFT effects are included in the ME but not in the parton shower, it is possible that the matching procedure could cause a mismatch by removing SMEFT effects that the parton shower cannot replace. It is expected that these effects are subdominant due to the additional momentum dependence introduced by SMEFT operators, as argued in Ref. [24]. For this reason, it is important to examine the DJR plots at various non-SM points within the SMEFT space to ensure that there are no signs of mismatches between the ME and parton shower contributions. These effects would be most relevant to investigate for operators involving gluons or light quarks. The generation and validation procedures for matched LO samples are described in more detail in Ref. [20].

Table 1: The list of EFT operators in the Warsaw basis [25] used in Sec. 5, categorized according to the main affected processes.

| Operator (CP even) | Multiboson | Studied in |
|---|---|---|
| $\epsilon^{IJK} W_\mu^{I\nu} W_\nu^{J\rho} W_\rho^{K\mu}$ | $\mathcal{O}_W$ | Sec. 5.1 |
| $\varphi^\dagger \tau^I \varphi W_{\mu\nu}^I B^{\mu\nu}$ | $\mathcal{O}_{HWB}$ | Sec. 5.2, 5.6 |
| $\varphi^\dagger \varphi W_{\mu\nu}^I W^{I\mu\nu}$ | $\mathcal{O}_{HW}$ | Sec. 5.2, 5.6 |
| $\varphi^\dagger \varphi B_{\mu\nu} B^{\mu\nu}$ | $\mathcal{O}_{HB}$ | Sec. 5.2, 5.6 |
| $(\varphi^\dagger \varphi)\Box(\varphi^\dagger \varphi)$ | $\mathcal{O}_{H\Box}$ | Sec. 5.2, 5.6 |
| $(\varphi^\dagger D_\mu \varphi)^*(\varphi^\dagger D_\mu \varphi)$ | $\mathcal{O}_{HD}$ | Sec. 5.2, 5.6 |
| **Operator (CP odd)** | **Multiboson** | **Section** |
| $\epsilon^{IJK} \tilde{W}_\mu^{I\nu} W_\nu^{J\rho} W_\rho^{K\mu}$ | $\mathcal{O}_{\tilde{W}}$ | Sec. 5.1 |
| $\varphi^\dagger \varphi \tilde{W}_{\mu\nu}^I W^{I\mu\nu}$ | $\mathcal{O}_{H\tilde{W}}$ | Sec. 5.2, 5.6 |
| $\varphi^\dagger \varphi \tilde{B}_{\mu\nu} B^{\mu\nu}$ | $\mathcal{O}_{H\tilde{B}}$ | Sec. 5.2, 5.6 |
| $\varphi^\dagger \tau^I \varphi \tilde{W}_{\mu\nu}^I B^{\mu\nu}$ | $\mathcal{O}_{H\tilde{W}B}$ | Sec. 5.2, 5.6 |
| **Operator (CP even)** | **Vector boson and quark** | **Section** |
| $i(\varphi^\dagger D_\mu \varphi - (D_\mu \varphi)^\dagger \varphi)(\bar{q}_p \gamma^\mu q_r)$ | $\mathcal{O}_{Hq}^{(1)}$ | Sec. 5.2, 5.6 |
| $i(\varphi^\dagger \tau^I D_\mu \varphi - (D_\mu \varphi)^\dagger \tau^I \varphi)(\bar{q}_p \tau^I \gamma^\mu q_r)$ | $\mathcal{O}_{Hq}^{(3)}$ | Sec. 5.2, 5.6 |
| $i(\varphi^\dagger D_\mu \varphi - (D_\mu \varphi)^\dagger \varphi)(\bar{u}_p \gamma^\mu u_r)$ | $\mathcal{O}_{Hu}$ | Sec. 5.2, 5.6 |
| $i(\varphi^\dagger D_\mu \varphi - (D_\mu \varphi)^\dagger \varphi)(\bar{d}_p \gamma^\mu d_r)$ | $\mathcal{O}_{Hd}$ | Sec. 5.2, 5.6 |
| $i(\tilde{\varphi}^\dagger D_\mu \varphi)(\bar{u}_p \gamma^\mu d_r)$ | $\mathcal{O}_{Hud}$ | Sec. 5.2, 5.6 |
| $(\bar{q}_p \sigma^{\mu\nu} u_r)\tau^I \tilde{\varphi} W_{\mu\nu}^I$ | $\mathcal{O}_{uW}$ | Sec. 5.2, 5.6 |
| $(\bar{q}_p \sigma^{\mu\nu} d_r)\tau^I \tilde{\varphi} W_{\mu\nu}^I$ | $\mathcal{O}_{dW}$ | Sec. 5.2, 5.6 |
| $(\bar{q}_p \sigma^{\mu\nu} u_r)\tilde{\varphi} B_{\mu\nu}$ | $\mathcal{O}_{uB}$ | Sec. 5.2, 5.6 |
| $(\bar{q}_p \sigma^{\mu\nu} d_r)\tilde{\varphi} B_{\mu\nu}$ | $\mathcal{O}_{dB}$ | Sec. 5.2, 5.6 |
| **Operator (CP even)** | **Top quark** | **Section** |
| $(\bar{q}_p \sigma^{\mu\nu} u_r)\tilde{\varphi} B_{\mu\nu}$ | $\mathcal{O}_{uB}$ | Sec. 5.4 |
| $(\bar{q}_p \sigma^{\mu\nu} u_r)\tau^I \tilde{\varphi} W_{\mu\nu}^I$ | $\mathcal{O}_{uW}$ | Sec. 5.4 |
| $(\bar{q}_i \sigma^{\mu\nu} T^A u_j)\tilde{\varphi} G_{\mu\nu}^A$ | $\mathcal{O}_{uG}^{(ij)}$ | Sec. 5.3 |
| $f^{ABC} G_\mu^{A\nu} G_\nu^{B\rho} G_\rho^{C\mu}$ | $\mathcal{O}_G$ | Sec. 5.3 |
| $\varphi^\dagger \varphi G_{\mu\nu}^A G^{A\mu\nu}$ | $\mathcal{O}_{\varphi G}$ | Sec. 5.3 |

Table 2: The list of EFT operators in the Warsaw basis [25] affecting processes with top quarks and which are used in Sec. 5.

| Operator (CP even) | Top quark (four fermion) | Section |
|---|---|---|
| $(\bar{q}_i\gamma^\mu q_j)(\bar{q}_k\gamma_\mu q_l)$ | $\mathcal{O}_{qq}^{1(ijkl)}$ | Sec. 5.3 |
| $(\bar{q}_i\gamma^\mu\tau^I q_j)(\bar{q}_k\gamma_\mu\tau^I q_l)$ | $\mathcal{O}_{qq}^{3(ijkl)}$ | Sec. 5.3 |
| $(\bar{q}_i\gamma^\mu q_j)(\bar{u}_k\gamma_\mu u_l)$ | $\mathcal{O}_{qu}^{1(ijkl)}$ | Sec. 5.3 |
| $(\bar{q}_i\gamma^\mu T^A q_j)(\bar{u}_k\gamma_\mu T^A u_l)$ | $\mathcal{O}_{qu}^{8(ijkl)}$ | Sec. 5.3 |
| $(\bar{q}_i\gamma^\mu q_j)(\bar{d}_k\gamma_\mu d_l)$ | $\mathcal{O}_{qd}^{1(ijkl)}$ | Sec. 5.3 |
| $(\bar{q}_i\gamma^\mu T^A q_j)(\bar{d}_k\gamma_\mu T^A d_l)$ | $\mathcal{O}_{qd}^{8(ijkl)}$ | Sec. 5.3 |
| $(\bar{u}_i\gamma^\mu u_j)(\bar{u}_k\gamma_\mu u_l)$ | $\mathcal{O}_{uu}^{(ijkl)}$ | Sec. 5.3 |
| $(\bar{u}_i\gamma^\mu u_j)(\bar{d}_k\gamma_\mu d_l)$ | $\mathcal{O}_{ud}^{1(ijkl)}$ | Sec. 5.3 |
| $(\bar{u}_i\gamma^\mu T^A u_j)(\bar{d}_k\gamma_\mu T^A d_l)$ | $\mathcal{O}_{ud}^{8(ijkl)}$ | Sec. 5.3 |
| $(\bar{q}_i u_j)\,\varepsilon\,(\bar{q}_k d_l)$ | $\mathcal{O}_{quqd}^{1(ijkl)}$ | Sec. 5.3 |
| $(\bar{q}_i T^A u_j)\,\varepsilon\,(\bar{q}_k T^A d_l)$ | $\mathcal{O}_{quqd}^{8(ijkl)}$ | Sec. 5.3 |

# 5 Simulation studies of EFT prediction methods

This section presents studies of the consistency of the different strategies for obtaining SMEFT predictions. Unless noted otherwise, both linear and quadratic terms are included. The vertical bars in the histogram correspond to $\sqrt{\sum_i w_i^2}$ where the index $i$ extends over all events in a bin and the event's weights are denoted by $w_i$. We list the operators used in the remainder of the work in Table 1 and Table 2.

## 5.1 Helicity and reweighting of predictions for the WZ process

Diboson production in proton-proton collisions is extremely important for studying the electroweak sector of the SM due to its sensitivity to the self-interaction of gauge bosons, probing anomalous effects in trilinear gauge couplings, and studying the interaction of massive vector bosons with quarks. Representative Feynman diagrams are shown in Fig. 1. In this section, we study the effects of the CP-even and -odd dimension-6 operators $\mathcal{O}_W$ and $\mathcal{O}_{\widetilde{W}}$ on the associated production of a W and Z boson, referred to as WZ production. These operators affect the triple gauge boson coupling, i.e., the interaction vertex involving three electroweak vector bosons, as shown in Fig. 1 (left). A detailed measurement of this process is performed by both the ATLAS and CMS Collaborations using Run-2 LHC data [26, 27].

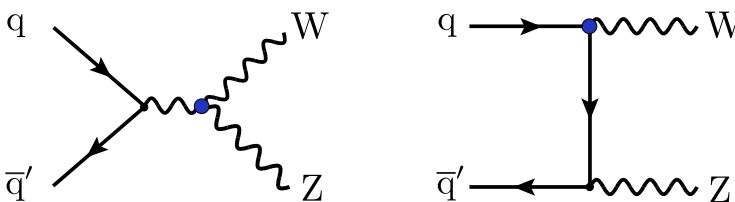

Figure 1: Feynman diagrams for WZ production with dimension-6 operators (blue markers) affecting the triple gauge boson vertex (left) and the vector boson coupling (right).

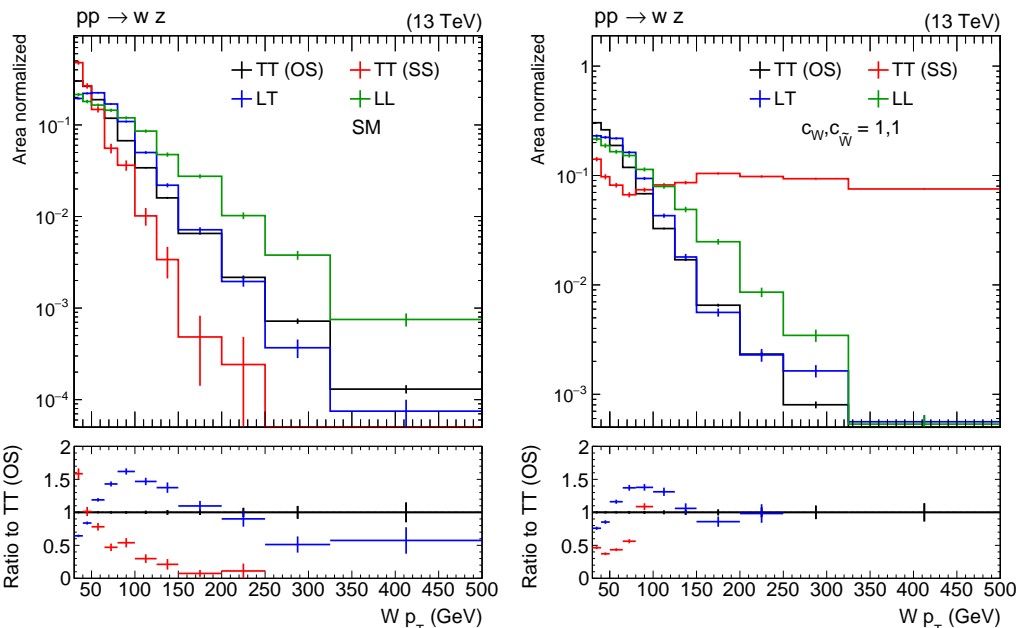

Figure 2: Helicity composition as a function of W boson $p_T$ at the SM point (left) and a BSM point with both $c_W$ and $c_{\tilde{W}}$ set to 1 (right). Here, L and T refer to longitudinal and transverse polarizations, respectively, whereas OS and SS refer to the opposite- and same-sign configurations, respectively. The helicity eigenstates are defined in the laboratory reference frame.

For the SM, the WZ process is generated at LO using MADGRAPH5_AMC@LO. The NNPDF3.1 NNLO PDF set [28] is used. The renormalization and factorization scales chosen are half of the sum of the transverse mass of final state particles. The SMEFT effects are simulated at LO using the SMEFTSIM v3.0 [29] model with the topU3l flavor scheme. Event samples are produced both at the SM point, i.e., setting all WCs to zero, and with non-zero values of the WCs for the operators considered. Several weights are stored for each event. These are computed using reweighting for the ME method [30], following two approaches: helicity-aware and helicity-ignorant reweightings. Ten million events are generated for each of the samples separately with helicity-aware and helicity-ignorant reweightings. For the generated samples, one million events are generated.

The WZ production in the SM is dominated by the helicity configuration where both bosons are transversely polarized with opposite helicities $(\pm, \mp)$, whereas the SMEFT operators considered here affect the configuration where both bosons have the same transverse helicities $(\pm, \pm)$ [31]. This is depicted in Fig. 2 as a function of W boson $p_T$ using the event samples produced for this study.

Both the SMEFT operators modify the W boson $p_T$ spectrum. Thus, it is used as an observable to compare between samples where the prediction at an EFT point is obtained using event weights and those produced directly at that particular EFT point, referred to as reweighted and generated predictions, respectively. The comparisons are shown in Fig. 3 for the case where $c_W$ and $c_{\tilde{W}}$ have a value of 1. The top row of Fig. 3 shows the comparison of W boson $p_T$ spectra, summed over all possible helicity configurations, in reweighted and generated samples for two choices of the reference point in reweighting: the SM point and a BSM point, where the WCs of both operators are set to 0.5.

For the SM reference point, the helicity-ignorant reweighting can reproduce the W boson $p_T$ spectrum predicted by the generated sample except at very high $p_T$ values, where the

sample size is small, but the helicity-aware reweighting fails. Both helicity-aware and -ignorant reweightings model the generated W boson $p_{\mathrm{T}}$ spectra very well once the BSM reference point is used in reweighting. The bottom row of Fig. 3 shows the same comparison as the top row but specifically for the helicity configuration affected by the SMEFT operators.

Here, it is evident that helicity-ignorant reweighting fails to model the $p_{\mathrm{T}}$ spectra for individual helicity configurations irrespective of the reference point chosen in reweighting, whereas the helicity-aware reweighting with a BSM reference point can model the W boson $p_{\mathrm{T}}$ spectrum for a specific helicity configuration affected by the SMEFT operators considered.

## 5.2 Helicity and reweighting of predictions for the ZH process

In this section, we study the modeling of SMEFT effects in Higgs production in association with a W or a Z boson, referred to as VH production. This particular Higgs production mode is extremely important for probing new physics, as its contribution becomes increasingly significant at high values of the Higgs or vector boson $p_{\mathrm{T}}$ [32]. The VH process has been measured by both the ATLAS and CMS Collaborations across different decay channels [33, 34].

The VH production is affected by a number of SMEFT operators at dimension 6 that modify the vector boson coupling to quarks, give rise to a four-point interaction, or modify the Higgs boson interaction with W or Z boson. We restrict to ZH production and focus on one vector coupling operator $\mathcal{O}_{Hq}^{(3)}$ and two HVV operators $\mathcal{O}_{HW}$ and $\mathcal{O}_{HW}$ that affect both WH and ZH productions. Representative Feynman diagrams are shown in Fig. 4. The final state with the Z boson decaying to leptons and the Higgs boson decaying to a pair of b quarks, as measured by both the ATLAS and CMS Collaborations [35–37], is considered. The $\mathcal{O}_{Hq}^{(3)}$ operator mainly affects the helicity configuration where the Z boson is longitudinally polarized, which is also dominant in the SM.

The HVV operators, on the other hand, also modify the interference of scattering amplitudes with transverse Z boson helicities. This affects the distribution of angular observables $\Theta$, $\hat{\theta}$, and $\hat{\phi}$, which are measured in the ZH rest frame [38], as depicted in Fig. 5.

The ZH production process, followed by the leptonic decay of the Z boson and the Higgs boson decay to a pair of bottom quarks, is generated at LO with up to one additional jet using MADGRAPH5_AMC@LO. The NNPDF3.1 NNLO PDF set [28] is used, with renormalization and factorization scales set to half of the sum of the transverse mass of the final state particles. The SMEFT effects are simulated at LO using the SMEFTSIM v3.0 [29] model with the topU3l flavor scheme. For each event, multiple weights are stored, computed via the ME reweighting using helicity-aware and helicity-ignorant reweighting. The SM point, where all WCs are set to zero, is used as the reference for reweighting.

Separate samples are generated by turning on one operator at a time with the following values: a) $c_{Hq}^{(3)} = 0.1$, b) $c_{HW} = 1$, c) $c_{H\tilde{W}} = 1$; in each case, all other WCs are set to zero. These are referred to as the generated samples. One million events are generated for each of the reweighted and generated samples. The particle-level Z boson $p_{\mathrm{T}}$ spectra predicted by reweighted samples are compared to those from the generated samples in Fig. 6. In this case, both helicity-aware and helicity-ignorant reweighting approaches accurately model the effects of $\mathcal{O}_{Hq}^{(3)}$ and $\mathcal{O}_{HW}$ on Z boson $p_{\mathrm{T}}$.

Next, we compare the distributions of the angles $\hat{\theta}$ and $\hat{\phi}$ depicted in Fig. 5 between reweighted and generated samples in Fig. 7 for $c_{HW}$ and $c_{H\tilde{W}}$. Both reweighting variants model the angular distributions obtained from the generated samples within statistical uncertainties. For the angle $\hat{\phi}$, we observe a $\cos 2\phi$ distribution for $c_{HW} = 1$, while for $c_{H\tilde{W}} = 1$, the distribution is modified due to the different interference terms relevant for CP-even and CP-odd gauge coupling operators. Therefore, $\hat{\phi}$ can be used to probe the CP nature of Higgs-to-vector boson interactions.

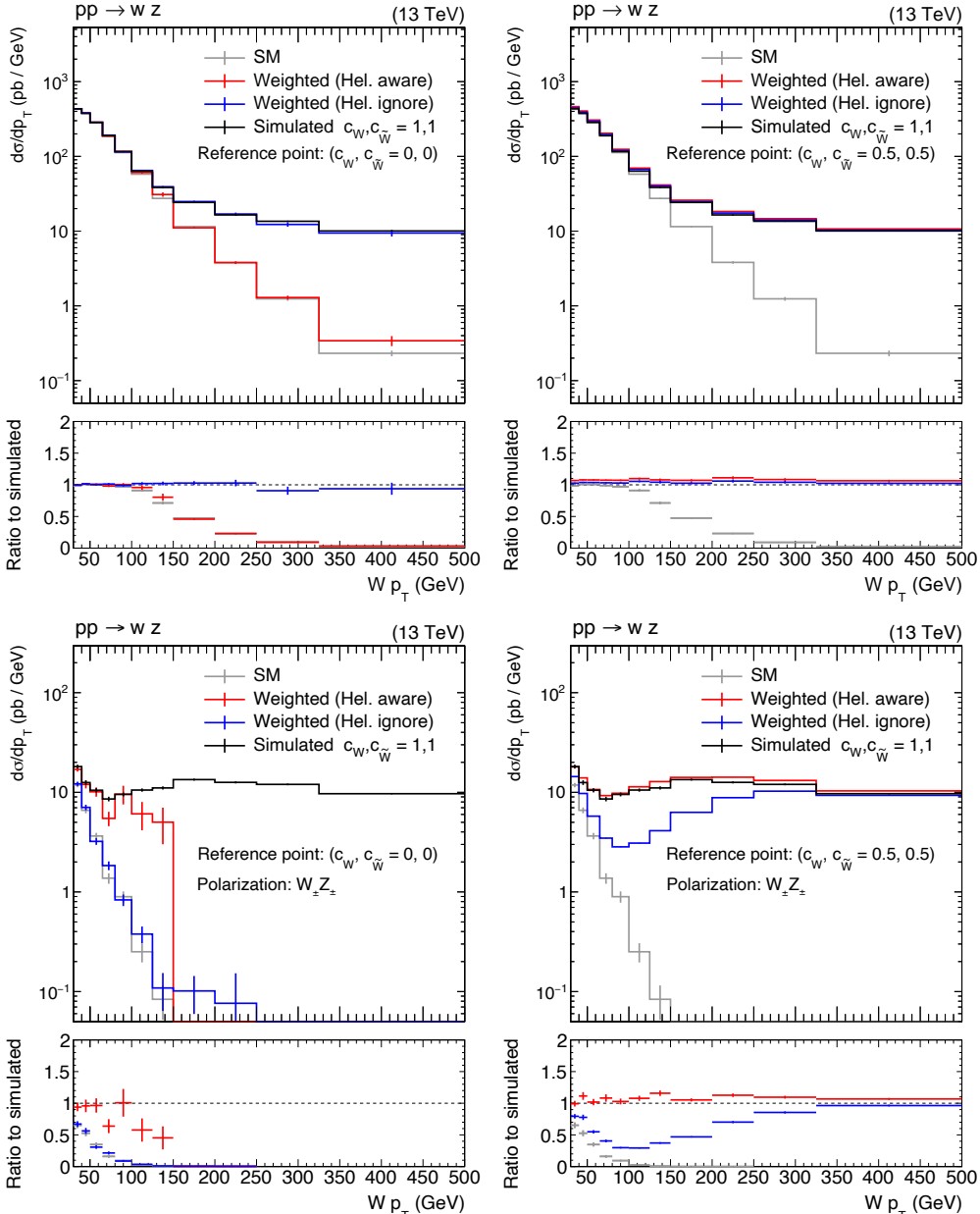

Figure 3: Comparison of W boson $p_T$ spectra between generated and reweighted (both helicity-aware and -ignorant variants) at a BSM point ($c_W, c_{\tilde{W}} = 1, 1$) for two reference points used in the reweighting: the SM point (left) and a BSM point with both $c_W$ and $c_{\tilde{W}}$ set to 0.5 (right). The upper row corresponds to the case where all helicity configurations are summed, and the lower row corresponds to only the same-sign transverse polarization configuration.

## 5.3 The $t\bar{t}$ process

In the SM, top-quark pair production ($t\bar{t}$) in proton-proton collisions occurs predominantly through gluon-gluon fusion ($\sim 90\%$), with quark-antiquark annihilation contributing the remaining $\sim 10\%$ at LO. At NLO in QCD and higher orders, channels with quark-gluon initial states also contribute. In this section, we include all operators that significantly impact $t\bar{t}$ production. We use the NNPDF3.1 NNLO PDF set [28] and the five-flavor scheme (5FS).

Table 2 lists two-heavy-two-light four-quark operators that can influence the $t\bar{t}$ process. Additionally, the operator $\mathcal{O}_{uG}^{(ij)}$ from Table 1 significantly affects the $t\bar{t}$ rate. Therefore, both the real and imaginary components of the WC for the $\mathcal{O}_{uG}^{(ij)}$ operator, denoted as ctGRe and ctGIm, are considered in this study.

In this work, we focus exclusively on the operators in the Warsaw basis that explicitly modify the couplings of the top quark with other SM fields. As a result, we exclude the $\mathcal{O}_G$ and $\mathcal{O}_{\varphi G}$ operators, which are well-constrained by processes not involving top quarks [39,40]. Representative Feynman diagrams are shown in Fig. 8.

The signal contribution is modeled at LO using the MADGRAPH5_AMC@LO event generator with the SMEFTSIM model to incorporate EFT effects. The default settings for helicity treatment in reweighting are used: at LO, helicity-aware reweighting is employed, while NLO samples are reweighted with helicity-ignorant settings. The definitions of the operators associated with all WCs are provided in [29].

As discussed in Sec. 1, the cross section (inclusive or differential) depends quadratically on the WCs. Each event weight is parameterized as a quadratic function of the WCs, as described in Eq. 8, by including sufficient reweighting points per event. The nominal $t\bar{t}$ sample is generated at a reference point in the WC parameter space, distant from the SM point, referred to as "LO (sample 1)," with the reference point denoted as Pt1.

To simulate EFT effects on $t\bar{t}$ production from quark-gluon initial states and more accurately predict distributions in the presence of extra jets, an additional sample is produced, including an extra final-state parton in the ME generation. This sample is also produced at Pt1

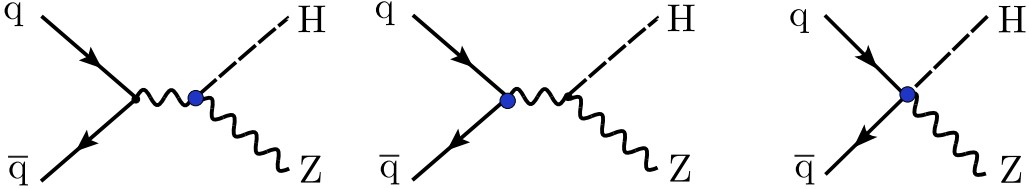

Figure 4: Feynman diagrams for ZH production with dimension-6 operators (blue markers) affecting the triple gauge boson vertex (left), the qqZ vertex (middle), and ZH production via an EFT four-point interaction (right).

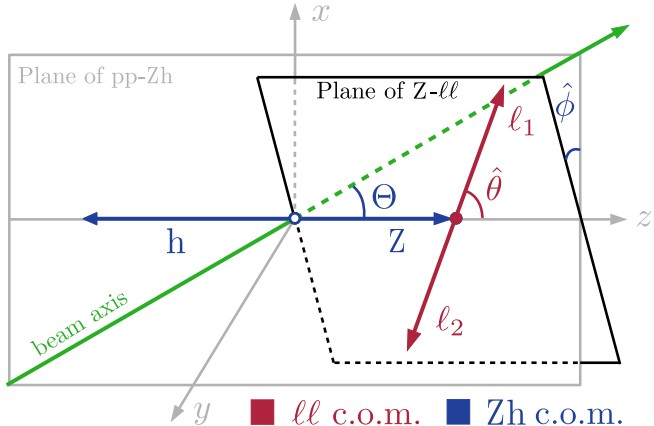

Figure 5: Decay planes and angles in $Z(\to \ell^+\ell^-)H(\to b\bar{b})$ production. The angle $\Theta$ and $\hat{\phi}$ are defined in the ZH rest frame, while $\hat{\theta}$ is defined in the Z boson rest frame.

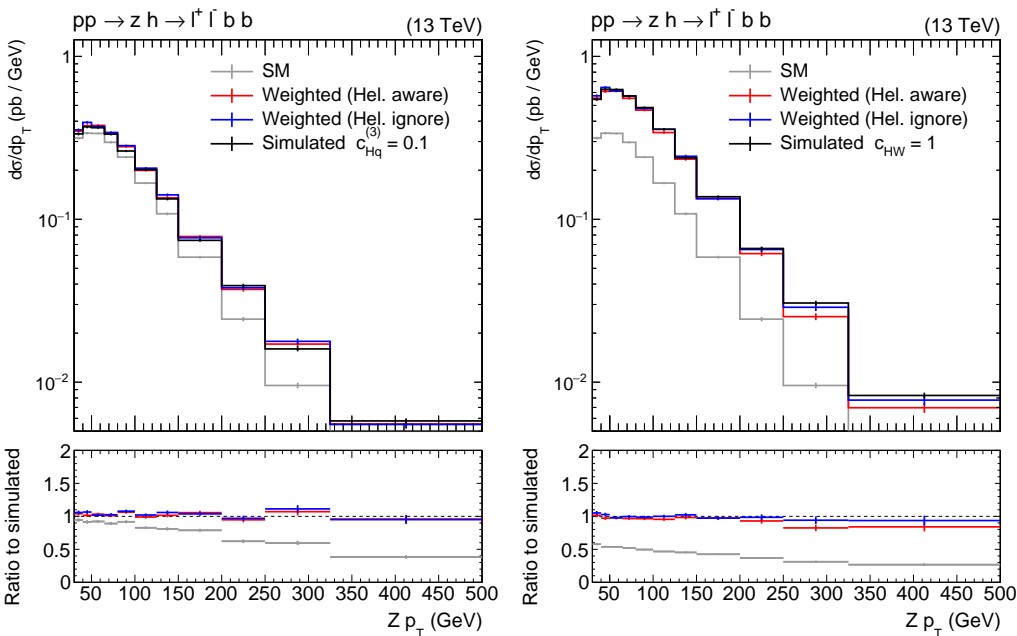

Figure 6: Comparison of Z boson $p_T$ spectra (absolute cross section per GeV on the y-axis) between generated and reweighted (both helicity-aware and -ignorant) ZH samples at $c_{Hq}^3 = 0.1$ (left) and $c_{HW} = 1$ (right). The helicity eigenstates are defined in the laboratory reference frame.

with MLM matching to the parton shower as described in Sec. 4 and is referred to as "LO+1 jet (sample 1)." Both $t\bar{t}$ and $t\bar{t}$ +1 jet samples include the dominant $t \to Wb$ decay, followed by leptonic W boson decays.

To ensure the generated sample can consistently reweight to other points in EFT parameter space, independent $t\bar{t}$ samples are generated at the following points in EFT space:

- SM point: all WCs set to zero.

- Dedicated point: All WCs set to zero except one which is set to -4,-2, 2, and 4 independently. The coefficient ctGRe is set to the smaller values -0.7, -0.4, -0.2, 0.2, 0.4, 0.7 because of its large effect on the cross section.

- Pt1: WCs set to $c_{tG}^{Im} = 0.7$, $c_{tG}^{Re} = 0.7$, $c_{Qj}^{38} = 9.0$, $c_{Qj}^{18} = 7.0$, $c_{Qu}^8 = 9.5$, $c_{Qd}^8 = 12.0$, $c_{tj}^8 = 7.0$, $c_{tu}^8 = 9.0$, $c_{td}^8 = 12.4$, $c_{Qj}^{31} = 3.0$, $c_{Qj}^{11} = 4.2$, $c_{Qu}^1 = 5.5$, $c_{Qd}^1 = 7.0$, $c_{tj}^1 = 4.4$, $c_{tu}^1 = 5.4$, $c_{td}^1 = 7.0$.

- Pt2: WCs set to $c_{tG}^{Im} = 1.0$, $c_{tG}^{Re} = 1.0$, $c_{Qj}^{38} = 3.0$, $c_{Qj}^{18} = 3.0$, $c_{Qu}^8 = 3.0$, $c_{Qd}^8 = 3.0$, $c_{tj}^8 = 3.0$, $c_{tu}^8 = 3.0$, $c_{td}^8 = 3.0$, $c_{Qj}^{31} = 3.0$, $c_{Qj}^{11} = 3.0$, $c_{Qu}^1 = 3.0$, $c_{Qd}^1 = 3.0$, $c_{tj}^1 = 3.0$, $c_{tu}^1 = 3.0$, $c_{td}^1 = 3.0$.

The samples produced at Pt2 are referred to as "LO (sample 2)" and "LO+1 jet (sample 2)." In Fig. 9 and Fig. 10, the relative SMEFT contributions to the $t\bar{t}$ inclusive cross section are shown for individual WCs. In each plot, the quadratic function extracted from LO (sample 1) and LO (sample 2) is compared to the cross section ratios calculated at the dedicated points. In general, there is good agreement between the reweighted cross section ratios and those calculated at the dedicated points in EFT space.

In addition to estimating the inclusive cross section, LO+1 jet (sample 1) should accurately predict differential distributions of various kinematic variables at different points in EFT space.

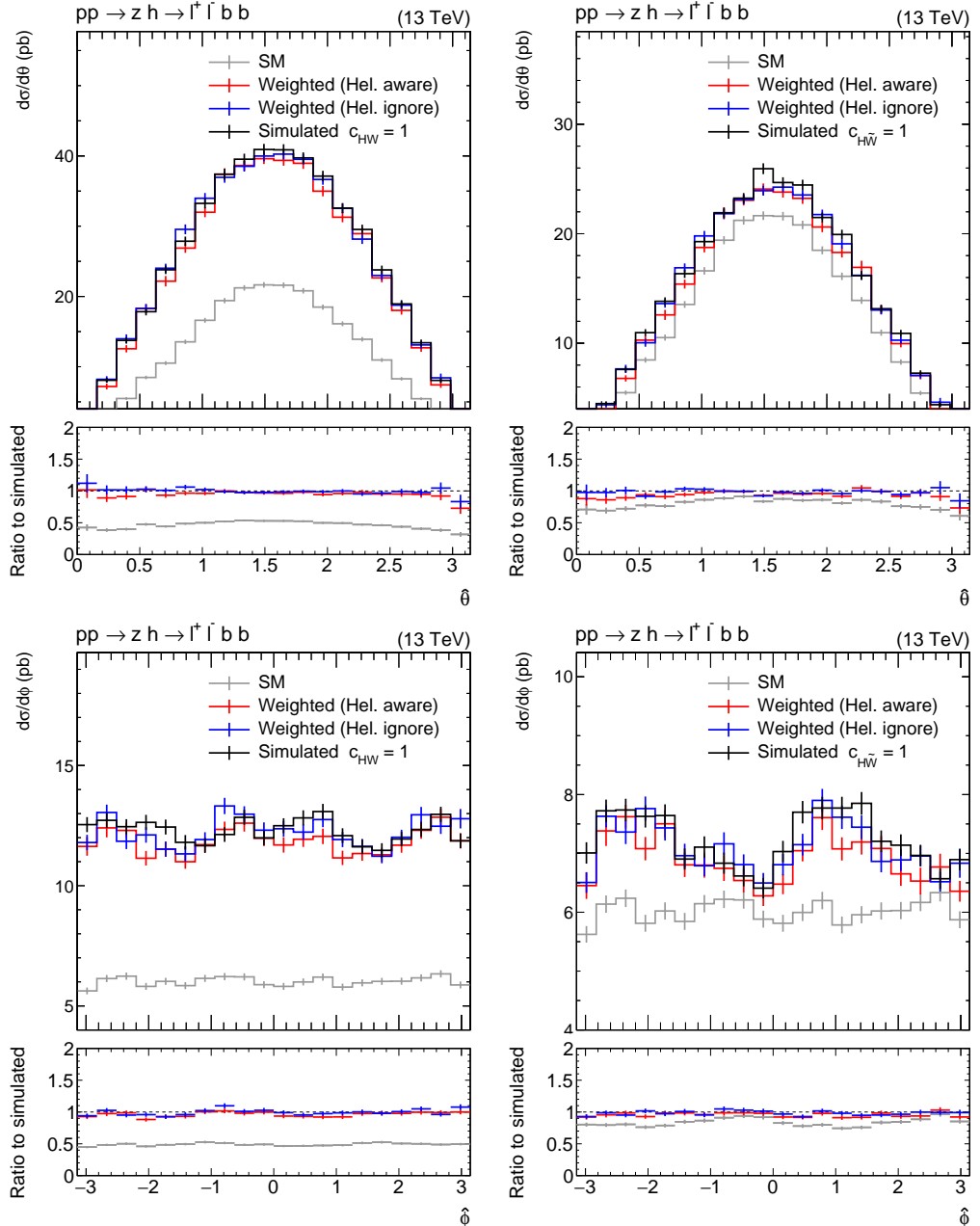

Figure 7: Comparison of $\hat{\theta}$ (top) and $\hat{\phi}$ (bottom) distributions (absolute cross section on the y-axis) between generated and reweighted (with both helicity-aware and -ignorant variants) ZH samples at $c_{HW} = 1$ (left) and $c_{H\tilde{W}} = 1$ (right). The angle $\hat{\phi}$ is defined in the ZH rest frame, while $\hat{\theta}$ is defined in the Z boson rest frame.

In Fig. 11, distributions of top quark $p_T$, leading lepton $p_T$, and $\Delta R$ between two leptons are shown for $t\bar{t}$ events with two leptons (electron or muon) with $p_T > 20$ GeV and $|\eta| < 2.5$, and at least two jets with $p_T > 20$ GeV and $|\eta| < 2.5$. The left column shows differential distributions for LO+1 jet (sample 1) and LO+1 jet (sample 2) reweighted to the SM point. The right column shows LO+1 jet (sample 1) and LO+1 jet (sample 2) reweighted to Pt2, the starting point of sample 2. These distributions demonstrate that the nominal sample can describe $t\bar{t}$ differential distributions across various points in EFT space, including the SM point.

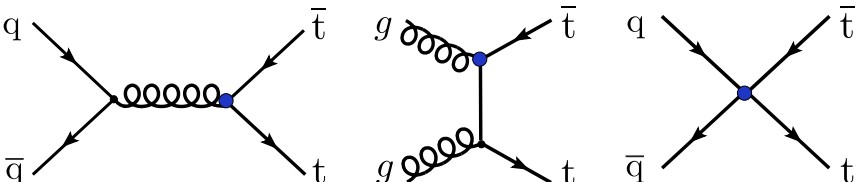

Figure 8: Feynman diagrams for $t\bar{t}$ production with dimension-6 operators (blue markers) affecting the interaction of the top quark and the gluon in gluon fusion production (left) and in t-channel production (middle), as well as via a four-fermion production vertex (right).

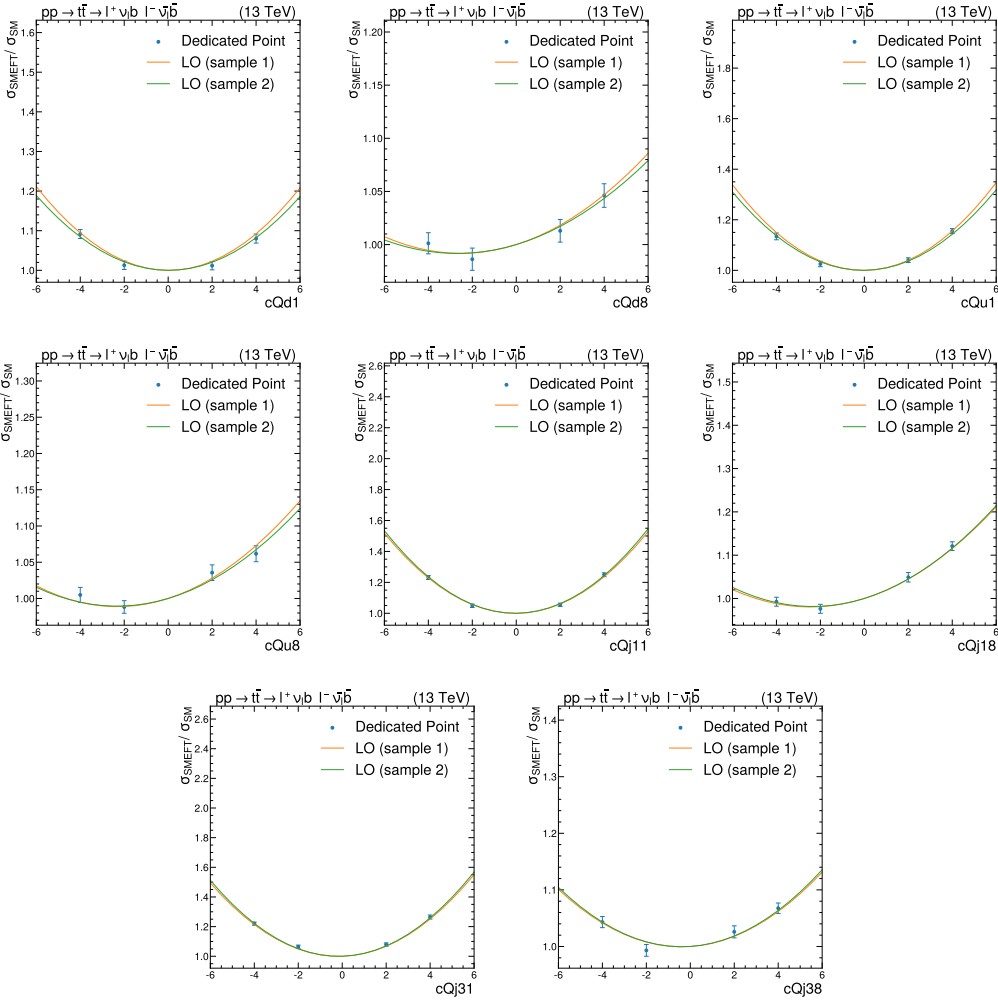

Figure 9: Relative modification of the $t\bar{t}$ total cross section, $\sigma_{\text{SMEFT}}/\sigma_{\text{SM}}$, induced by the presence of the SMEFT operators. Solid curves show the quadratic dependency of the relative cross section to the WC values obtained from reweighting sample 1 and sample 2 in EFT space. Blue points are obtained from dedicated simulations.

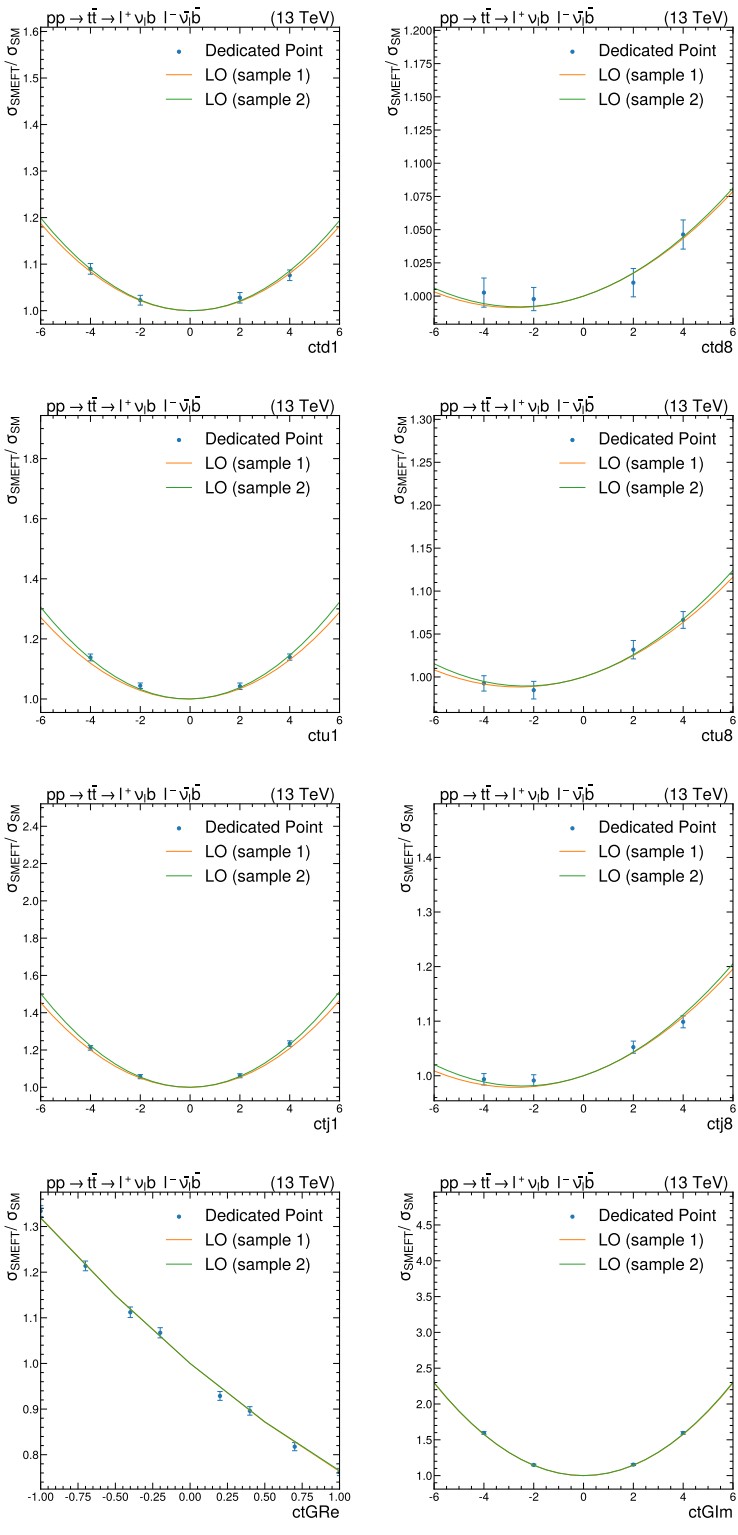

Figure 10: Relative modification of the $t\bar{t}$ total cross section, $\sigma_{\text{SMEFT}}/\sigma_{\text{SM}}$, induced by the presence of the SMEFT operators. Solid curves show the quadratic dependency of the relative cross section to the WC values obtained from reweighting sample 1 and sample 2 in EFT space. Blue points are obtained from dedicated simulations.

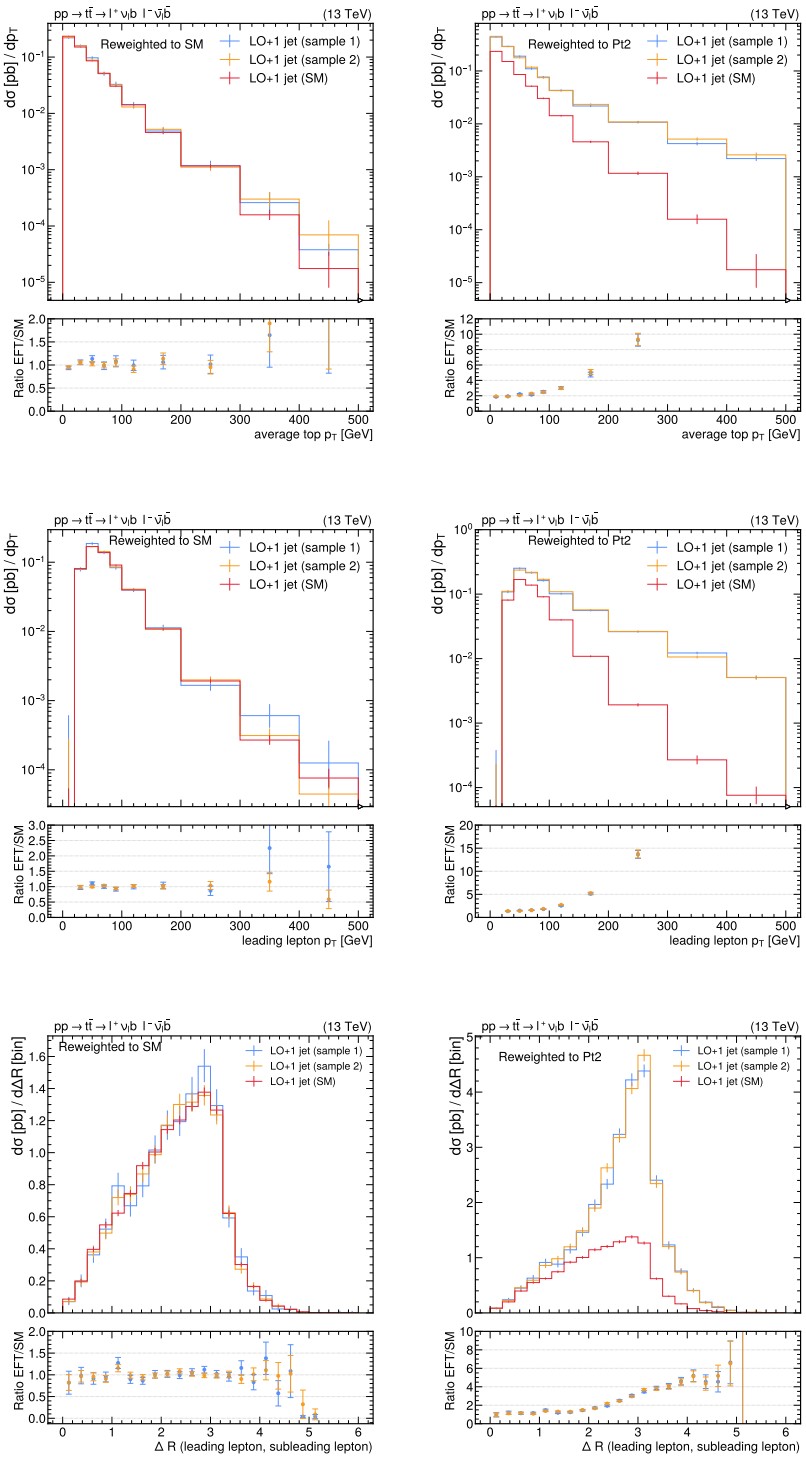

Figure 11: Differential distributions for t$\bar{t}$ +1jet process with respect to the top quark average $p_T$ (top), leading lepton $p_T$ (middle) and the $\Delta R$ angular distance between the leading and the sub-leading lepton(bottom). Differential distributions obtained from reweighting both sample 1 and sample 2 to the SM point (left) and reweighting sample 1 to Pt2 (right).

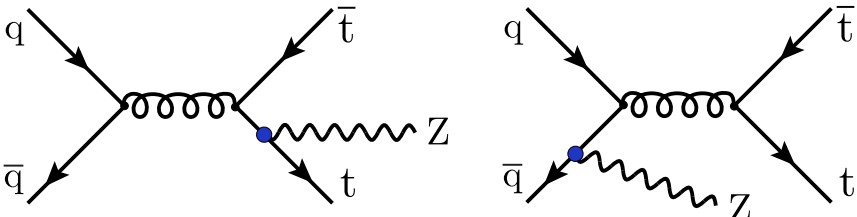

Figure 12: Feynman diagrams for $t\bar{t}Z$ production with dimension-6 operators (blue markers) affecting the interaction of the top quark and the Z boson (left) and with first- and second-generation fermions (right).

## 5.4 Studies in the $t\bar{t}Z$ process at NLO

In this section, we analyze the EFT reweighting performance using the production of a top-antitop pair in association with a Z boson emission ($t\bar{t}Z$ process) as a benchmark. This process has been extensively studied by both ATLAS [41,42] and CMS [43,44], showing high sensitivity to possible EFT effects from operators that modify the interaction of quarks with the Z boson. Representative Feynman diagrams are shown in Fig. 12.

We focus on the $\mathcal{O}_{tZ}$ operator, which modifies the interaction between the top quark and the Z boson, defined as $-\sin\theta_W \mathcal{O}_{uB} + \cos\theta_W \mathcal{O}_{uW}$, with $\mathcal{O}_{uB}$ and $\mathcal{O}_{uW}$ from Table 1.

To simulate the $t\bar{t}Z$ process, we use MADGRAPH5_AMC@NLO, incorporating EFT effects with the SMEFTSIM [29] and SMEFT@NLO [45] models. Events are generated at NLO in QCD using the SMEFT@NLO model and at LO plus one additional parton using the SMEFTSIM model, with parton showering modeled using PYTHIA8 [46]. For the LO samples, particular attention is given to the matching procedure, as discussed in Sec. 4. Both simulations are performed in the 5-flavor scheme (5FS), with renormalization and factorization scales set to half the sum of the transverse masses. Events are generated assuming the SM and assuming $c_{tZ} = 1$ in the SMEFT@NLO convention. For the SMEFTSIM prediction, the conversion of WCs provided in Ref. [29] is employed.

In Fig. 13 and Fig. 14, the distributions of the top quark and Z boson transverse momentum are shown for the SM and different EFT prediction strategies. The left plots display distributions from the LO simulation, while the right plots present the NLO results. It is not possible to separate linear and quadratic EFT contributions in the NLO simulation in MADGRAPH5_AMC@NLO v2, so only reweighted and separate simulation distributions are shown. This separation is possible in MADGRAPH5_AMC@NLO v3.

In the central panels of Fig. 13, the ratio between the EFT distributions and the SM is shown, highlighting increased sensitivity to EFT effects in the tail of the top quark transverse momentum distribution. The bottom panels display the ratio between the different EFT generation strategies, demonstrating good agreement across the entire spectrum.

The Z boson transverse momentum distribution, shown in Fig. 14, exhibits greater sensitivity to EFT effects compared to the top $p_T$ distribution. However, the agreement between the reweighted simulation and the other two EFT prediction strategies worsens at high Z $p_T$ in the LO results. This discrepancy arises from employing helicity-aware reweighting using the SM as a reference point. In the NLO results, where only helicity-ignorant reweighting is possible, there remains good agreement between the reweighted and direct simulations.

One drawback of the reweighting simulation is the presence of statistical fluctuations in poorly populated phase space due to large event weights. This is evident in Fig. 13 (left) and Fig. 14, where large statistical fluctuations are observed in the reweighted distribution at high Z and top transverse momentum.

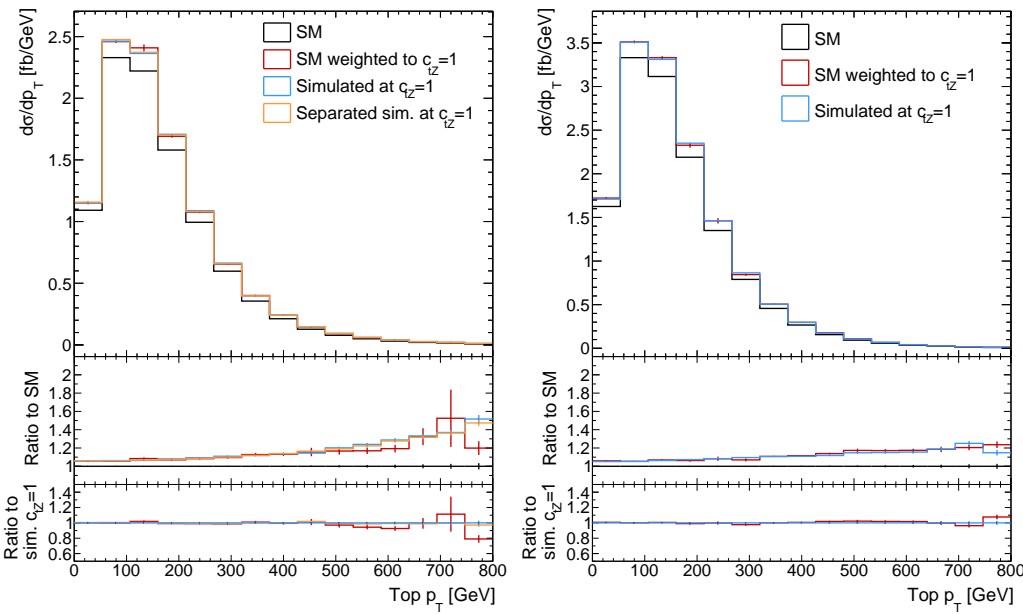

Figure 13: Top $p_T$ distribution simulated at LO + 1 jet (left) and at NLO (right). The SM distribution is shown in black, while the other lines represent the different available simulation methods to compute EFT predictions. The first ratio plot highlights the sensitivity to the EFT effects, while the second ratio plot shows the agreement between the direct simulation, the reweighted simulation, and the separate simulation (only for LO + 1 jet).

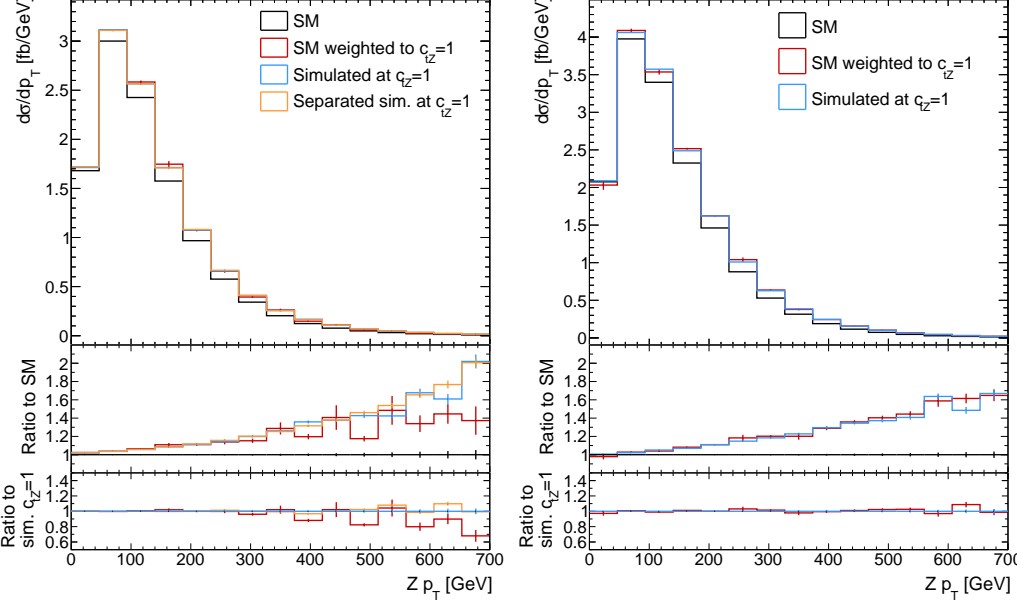

Figure 14: Z $p_T$ distribution simulated at LO + 1 jet (left) and at NLO (right). The SM distribution is shown in black, while the other lines represent the different available simulation methods to compute EFT predictions. The first ratio plot highlights the sensitivity to the EFT effects, while the second ratio plot shows the agreement between the direct simulation, the reweighted simulation, and the separate simulation (only for LO + 1 jet).

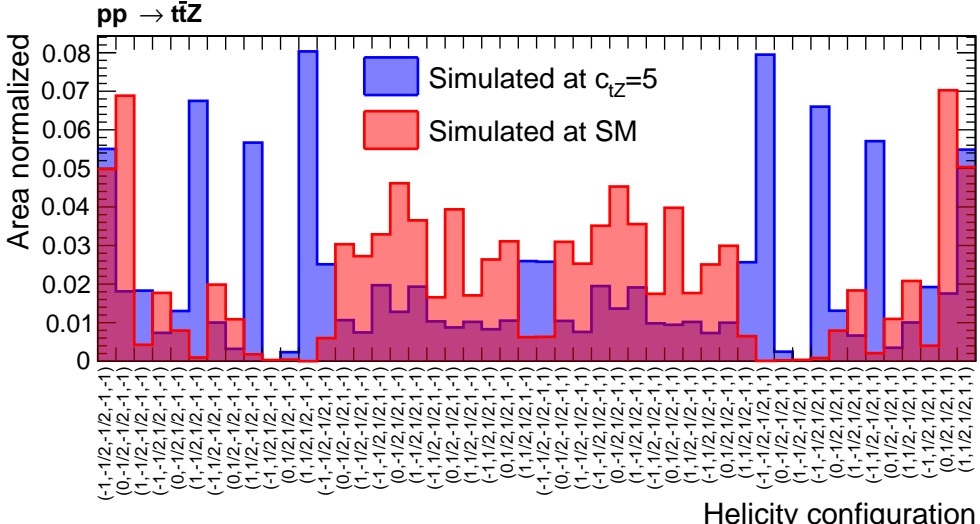

Figure 15: Distribution of different helicity configurations in t$\bar{\text{t}}$Z events generated at SM and $c_{\text{tZ}} = 5$ in the partonic center-of-mass frame. Bins are labeled as $(h_{\text{Z}}, h_{\text{t}}, h_{\bar{\text{t}}}, h_{\text{g}_1}, h_{\text{g}_2})$, where $h_i$ denotes the helicity of each particle. The plot shows that certain helicity configurations, such as those with $h_{\text{g}_1} = h_{\text{g}_2}$, are suppressed in the SM, meaning helicity-aware reweighting cannot populate those phase space regions.

## 5.5 Helicity aware and ignorant reweighting in the t$\bar{\text{t}}$Z process

In this section, we use the t$\bar{\text{t}}$Z process, introduced in Sec. 5.4, to showcase the benefits and limitations of the helicity-ignorant reweighting approach. To this end, we generate t$\bar{\text{t}}$Z events using MADGRAPH5_AMC@LO and simulate EFT effects from the $\mathcal{O}_{\text{tZ}}$ operator.

The $\mathcal{O}_{\text{tZ}}$ operator, defined as $-\sin\theta_W \mathcal{O}_{uB} + \cos\theta_W \mathcal{O}_{uW}$, introduces t$\bar{\text{t}}$Z vertices with helicity configurations absent in the SM, making the choice between helicity-ignorant and helicity-aware reweighting particularly relevant. This is illustrated in Fig. 15, which shows the distribution of the spins of the different partons involved in t$\bar{\text{t}}$Z events. We consider events generated under the SM and with $c_{\text{tZ}} = 5$.

The figure demonstrates that certain helicity configurations naturally occurring in the BSM scenario are either suppressed or nonexistent in the SM. Consequently, it is impossible to use the helicity-aware method to reweight SM samples to reproduce this specific BSM scenario: the phase space regions spanned by these helicity configurations will not be populated by SM samples.

This limitation is further demonstrated in Fig. 16, which shows the inclusive t$\bar{\text{t}}$Z cross-section dependence as a function of $c_{\text{tZ}}$. This dependence is computed using three independent samples with the same number of events. Two samples are generated assuming the SM, and a third is generated at $c_{\text{tZ}} = 5$. These samples are then reweighted to $c_{\text{tZ}} = 0, -1, 1$ to obtain the inclusive t$\bar{\text{t}}$Z cross-section for those values. From these points, the dependence is interpolated using a second-order polynomial, with the coefficients shown in Fig. 16. The $c_{\text{tZ}} = 5$ sample serves as the reference, expected to populate the full kinematic phase space more effectively. We use the helicity-aware and ignorant reweighting for each of the SM samples.

The trend predicted by the helicity-ignorant reweighting approach aligns well with the reference value. In contrast, the helicity-aware approach predicts a significantly smaller quadratic term and exhibits larger statistical uncertainties compared to the other methods. This discrepancy arises from two key factors. First, the smaller value predicted by the helicity-aware method is because it cannot populate phase space regions that are forbidden in the SM but al-

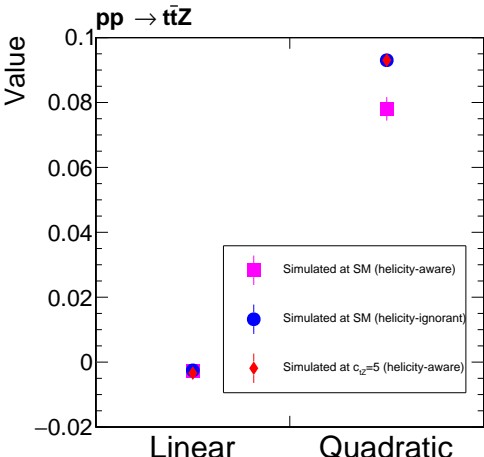

Figure 16: Quadratic parametrization of the $t\bar{t}Z$ cross-section as a function of $c_{tZ}$, obtained by reweighting samples generated at the SM and $c_{tZ} = 5$ using the helicity-ignorant and helicity-aware methods.

lowed when $c_{tZ} \neq 0$. Second, the larger uncertainties are due to a degradation in the statistical power of the weighted sample, resulting from large event weights in regions of phase space that are suppressed, though not forbidden, in the SM. These results highlight two advantages of the helicity-ignorant reweighting: it can reweight SM samples to certain BSM scenarios and increase the statistical power of weighted samples.

Although helicity-ignorant reweighting more efficiently populates the kinematic phase space, the reweighted samples do not necessarily reproduce the helicity configurations of the target scenario. The helicity of the produced partons is not directly measurable, but it can introduce correlations among the kinematic variables of the final state particles. In analyses where these correlations are relevant, it is crucial to keep track of the helicity of the different particles.

To study this effect, we consider two different methods for modeling the decay of the produced partons. In both scenarios, we generate samples with $c_{tZ} = 5$ and reweight them to the SM, alongside samples produced directly at the SM. In one scenario, we use MADSPIN [47] to model the decays of the top quarks and Z boson, computing weights based on the top quarks and Z boson before decay. In the other scenario, the decays are modeled using the MADGRAPH5_AMC@LO decay syntax, and the weights are computed based on the particles produced in the decays of the top quarks and Z boson. For this process, we expect shortcomings in the helicity-ignorant reweighting only in the first scenario, since in the latter case, the reweighting is performed based on the final state particles, making changes in helicity irrelevant to observable effects.

In Fig. 17, we show the $\Delta\phi$ distribution between the two leptons produced in the Z boson decay. The plot demonstrates that only the MADGRAPH5_AMC@LO decay model produces consistent results, while Madspin introduces artificial trends. These trends arise because Madspin implicitly samples from $\mathcal{M}_{prod+decay}/\mathcal{M}_{prod}$, the ratio of the production-plus-decay over production MEs, which is computed at the reference point. By default, Madspin does not recompute this ratio for reweighted events, introducing artificial biases when reweighting to a scenario where $\mathcal{M}_{prod+decay}/\mathcal{M}_{prod}$ differs from the reference point.

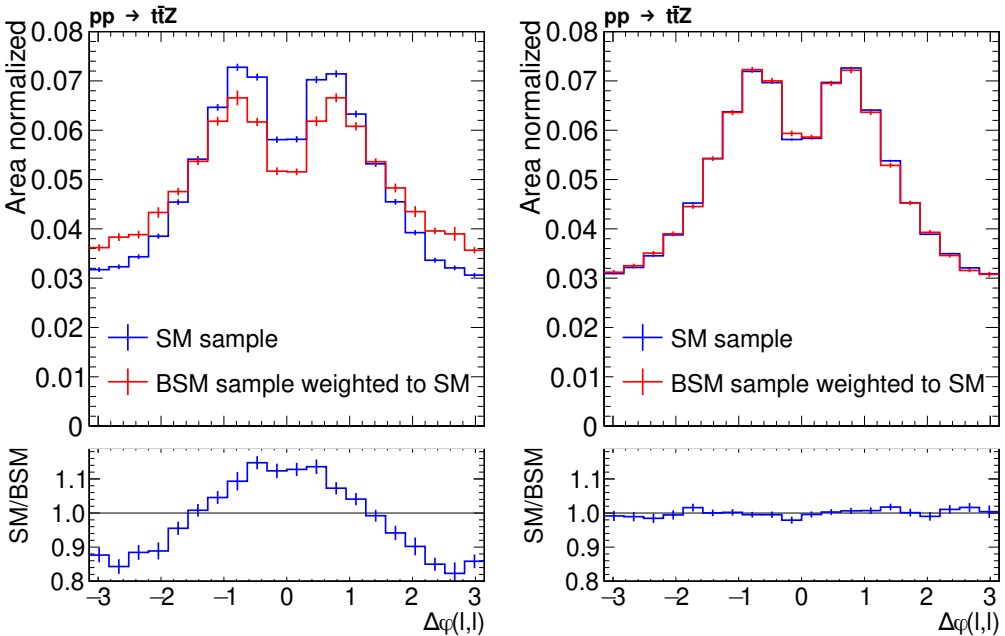

Figure 17: Distribution of $\Delta\phi$ between the two leptons produced in Z boson decay, for samples where the top quarks and Z boson are decayed using Madspin (left) and the MADGRAPH5_AMC@LO decay syntax (right). Samples generated at the SM and $c_{tZ} = 5$ are both reweighted to the SM.

## 5.6 The VBF H process

Vector boson fusion (VBF) is one of the dominant Higgs boson production mechanisms at the LHC. This process has been measured by the ATLAS and CMS experiments across various Higgs boson decay channels [33,34]. These measurements rely on the presence of two forward (high $|\eta|$) quark-initiated jets to distinguish VBF from other Higgs production modes.

Dimension-6 operators in the Standard Model effective field theory (SMEFT), including CP-odd ones, can modify the VBF process in several ways by:

1. altering the total cross section,

2. introducing anomalous couplings between the Higgs and vector bosons, as shown in Fig. 18 (left),

3. introducing anomalous couplings between the quarks and vector bosons (Fig. 18, middle) or HVqq contact interactions (Fig. 18, right).

Operators that affect Higgs boson decays are not considered in this study.

An SM VBF Higgs sample is generated at LO using MADGRAPH5_AMC@LO v2.9.13 [17] and the NNPDF3.1 PDF set [48]. The renormalization and factorization scales are set dynamically,

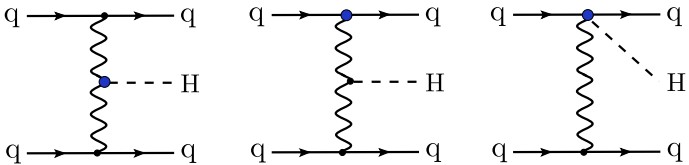

Figure 18: Feynman diagrams representing VBF Higgs boson production with EFT contributions (blue markers) to the HVV interaction (left), the Vqq interaction (middle), and a VHqq contact interaction (right).

Table 3: VBF cross sections calculated by MADGRAPH_AMC@LO from $10^6$ simulated events.

| Cross section [pb] | Direct simulation | Reweighted | SM + Lin. + Quad. |
|---|---|---|---|
| SM | $3.637 \pm 0.027$ | — | — |
| $c_{H\square} = 1$ | $4.091 \pm 0.031$ | $4.085 \pm 0.058$ | $4.091 \pm 0.031$ |
| $c_{HW} = 1$ | $3.563 \pm 0.032$ | $3.558 \pm 0.068$ | $3.567 \pm 0.036$ |
| $\tilde{c}_{HW} = 1$ | $3.762 \pm 0.022$ | $3.809 \pm 0.068$ | $3.781 \pm 0.032$ |
| $c_{Hj^{(1)}} = 1$ | $3.793 \pm 0.026$ | $3.815 \pm 0.075$ | $3.812 \pm 0.032$ |
| $c_{Hj^{(3)}} = 1$ | $2.774 \pm 0.023$ | $2.705 \pm 0.052$ | $2.759 \pm 0.047$ |

using the default values in MADGRAPH5_AMC@LO, which correspond to the transverse mass of the $2 \rightarrow 2$ system resulting from $k_{\mathrm{T}}$ clustering. The SMEFTSIM framework [29], with the topU3l model and $m_W$ input parameter scheme, is employed to provide per-event weights relative to the SM, accounting for EFT contributions (both linear and quadratic) from each operator listed in Table 1. The EFT scale $\Lambda$ is fixed at 1 TeV, and only Feynman diagrams with single-operator insertions are considered. The reweighting procedure used for this sample is helicity-aware.

It is important to note that since the $\mathcal{O}_{Hud}$ operator induces a right-handed charged current, its effects are not captured in this simulated sample.

Five EFT scenarios are selected as representative examples of the three classes enumerated above: $c_{H\square} = 1$, $c_{HW} = 1$, $\tilde{c}_{HW} = 1$, $c_{Hq^{(1)}} = 1$, and $c_{Hq^{(3)}} = 1$. In each case, only the listed WC is non-zero. In addition to the reweighted SM sample, each of these five EFT points is simulated directly up to quadratic order. For each EFT point, two additional samples are generated: one includes only the linear term, and the other only the quadratic term. The sum of the Standard Model (SM), linear-only, and quadratic-only samples is expected to reproduce the full sample. The cross section of each EFT sample is listed in Table 3. Good agreement is observed between the various simulation methods.

Figure 19(a) shows the distribution of the Higgs boson $p_{\mathrm{T}}$ for the $c_{H\square} = 1$ scenario, for each of the simulated samples. The overall cross section is enhanced compared to the SM, as indicated in Table 3. Note that the upper limit of the $p_{\mathrm{T}}$ distribution (700 GeV) may extend beyond the range of EFT validity. However, as this study aims to validate the performance of different simulation strategies, the range of EFT validity is not considered. Figure 19(b) shows the angular separation $\Delta\eta$ between the spectator quarks, which is similar in shape to the SM expectation.

Figure 20(a) shows the Higgs boson $p_{\mathrm{T}}$ distribution for the $c_{HW} = 1$ scenario. Compared to the SM, a deficit is observed below 50 GeV, and an enhancement is seen above approximately 150 GeV. Figure 20(b) shows the azimuthal separation $\Delta\phi$ between the spectator quarks for the $c_{HW} = 1$ scenario. This distribution is significantly modified by the EFT operator: while the SM expectation shows a preference for large angular separations, the distribution for $c_{HW} = 1$ is nearly flat. This occurs because the $\mathcal{O}_{HW}$ operator affects only the transverse amplitude $qV_T \rightarrow qH$ in VBF Higgs production [49].

Figure 21 shows the distribution of the Higgs boson $p_{\mathrm{T}}$ (a) and the azimuthal separation $\Delta\phi$ between the spectator quarks (b) for the $\tilde{c}_{HW} = 1$ scenario. Compared to the SM expectation, an enhancement in $p_{\mathrm{T}}$ is observed above approximately 150 GeV. The $\Delta\phi$ distribution is also slightly modified, though the effect is much weaker than that seen in Fig. 20b for the conjugate operator, as the linear term, which models the interference between the SM and the EFT scenario, is small.

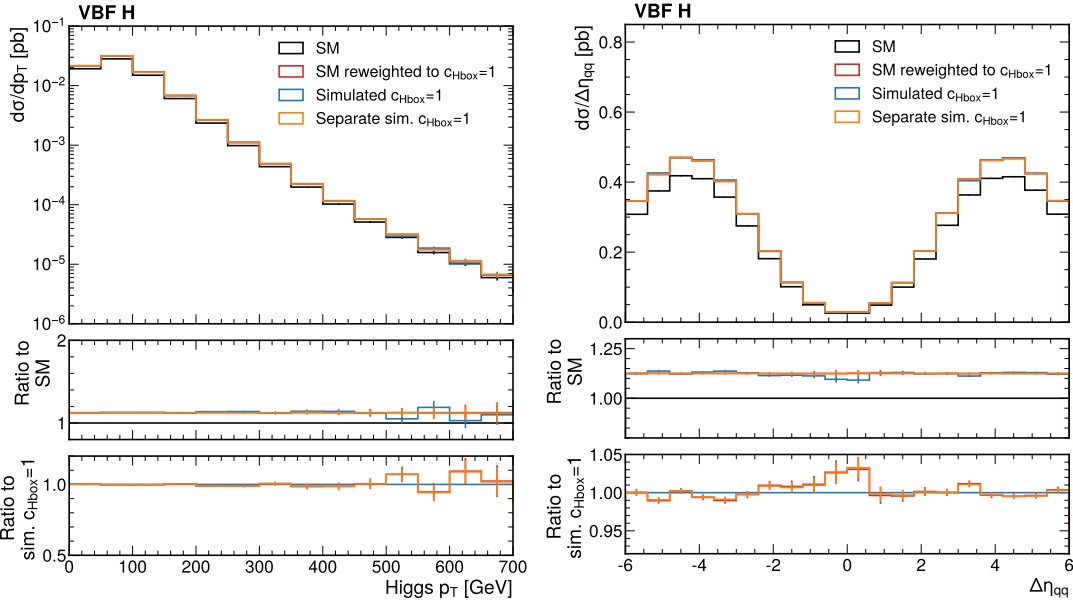

Figure 19: Distribution of (a) the Higgs boson $p_T$ and (b) the angular separation $\Delta\eta$ between the spectator quarks, for the $c_{H\square} = 1$ scenario. The SM expectation is shown in black. The prediction for $c_{H\square} = 1$ obtained by reweighting the SM point is shown in red. The direct simulation of $c_{H\square} = 1$ is shown in blue, and the sum of SM, linear-only, and quadratic-only is shown in orange. The middle panel shows the ratio to the SM, and the lower panel shows the ratio to the directly simulated sample.

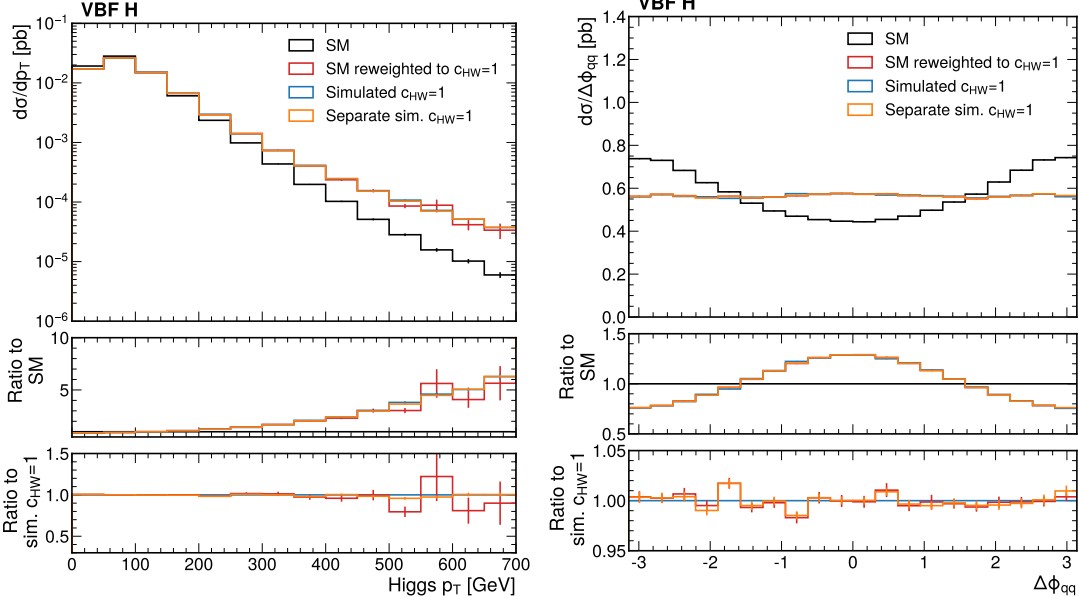

Figure 20: Distribution of (a) the Higgs boson $p_T$ and (b) the angular separation $\Delta\phi$ between the spectator quarks, for the $c_{HW} = 1$ scenario. The SM expectation is shown in black. The prediction for $c_{HW} = 1$ obtained by reweighting the SM point is shown in red. The direct simulation of $c_{HW} = 1$ is shown in blue, and the sum of SM, linear-only, and quadratic-only is shown in orange. The middle panel shows the ratio to the SM, and the lower panel shows the ratio to the directly simulated sample.

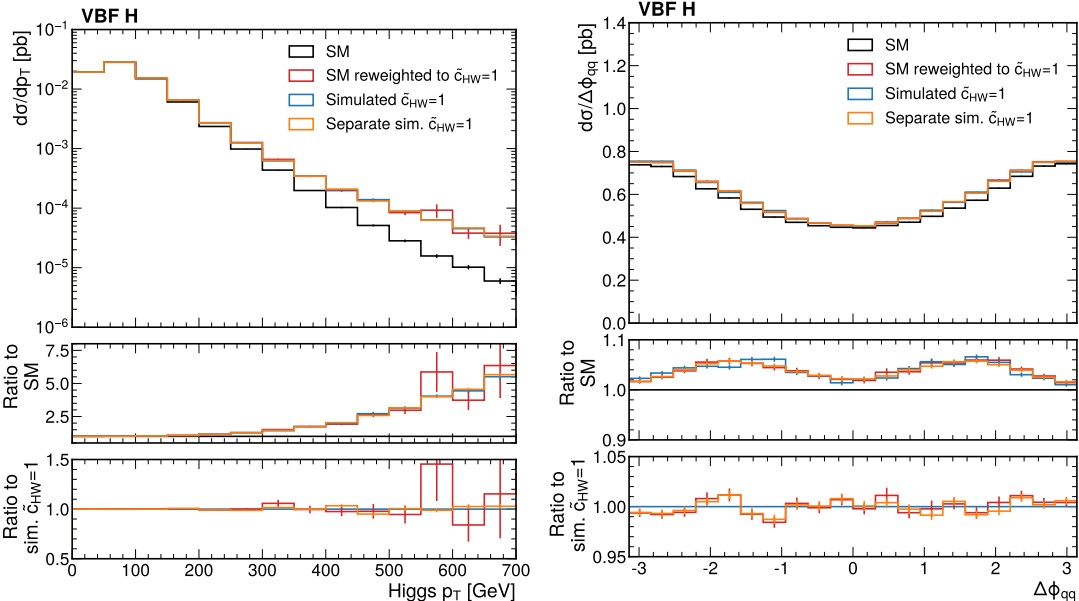

Figure 21: Distribution of (a) the Higgs boson $p_T$ and (b) the azimuthal separation $\Delta\phi$ between the spectator quarks for the $\tilde{c}_{HW} = 1$ scenario. The SM expectation is shown in black. The prediction for $\tilde{c}_{HW} = 1$ obtained by reweighting the SM point is shown in red. The direct simulation of $\tilde{c}_{HW} = 1$ is shown in blue, and the sum of SM, linear-only, and quadratic-only contributions is shown in orange. The middle panel shows the ratio to the SM, and the lower panel shows the ratio to the directly simulated sample.

Figure 22 shows the distribution of the Higgs boson $p_T$ (a) and the angular separation $\Delta\eta$ between the spectator quarks (b) for the $c_{Hj^{(1)}} = 1$ scenario. Compared to the SM expectation, an enhancement in $p_T$ is observed above approximately 150 GeV. It should be noted that the value of $c_{Hj^{(1)}} = 1$ is large compared to existing constraints [50], and the large deviation from the SM at high $p_T$ may extend beyond the range of EFT validity. A small enhancement with respect to the SM is also observed for $|\Delta\eta| \sim 0$.

Figure 23 shows the distribution of Higgs boson $p_T$ (a) and the angular separation $\Delta\eta$ between the spectator quarks (b) for the $c_{Hj^{(3)}} = 1$ scenario. Similar to $c_{Hj^{(1)}}$, the value of $c_{Hj^{(3)}} = 1$ is large compared to existing constraints [50]. Compared to the SM expectation, an overall lower cross section is observed, with a deficit in the range $100 < p_T < 300$ GeV. A small deficit is also observed for $|\Delta\eta| \sim 0$.

The comparison of these simulated EFT samples indicates good agreement between the predictions obtained by reweighting the SM sample, directly simulating, and combining separately generated SM, linear, and quadratic components. In regions where the SM sample contains a limited number of events, such as at the highest $p_T$ and at small $|\Delta\eta|$, some fluctuations and large uncertainties are observed in the reweighted spectrum.

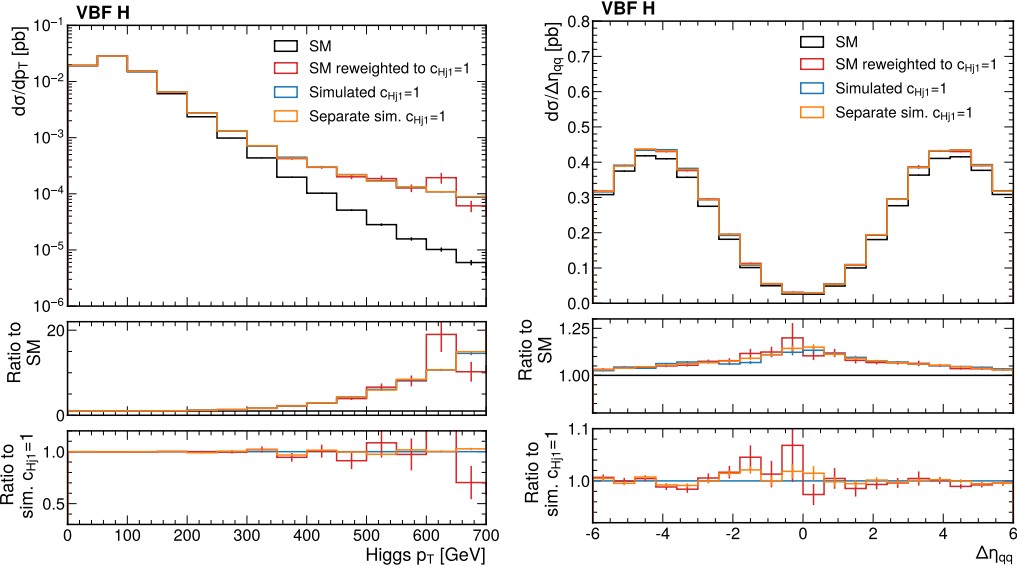

Figure 22: Distribution of (a) the Higgs boson $p_T$ and (b) the angular separation $\Delta\eta$ between the spectator quarks for the $c_{Hj^{(1)}} = 1$ scenario. The SM expectation is shown in black. The prediction for $c_{Hj^{(1)}} = 1$ obtained by reweighting the SM point is shown in red. The direct simulation of $c_{Hj^{(1)}} = 1$ is shown in blue, and the sum of SM, linear-only, and quadratic-only contributions is shown in orange. The middle panel shows the ratio to the SM, and the lower panel shows the ratio to the directly simulated sample.

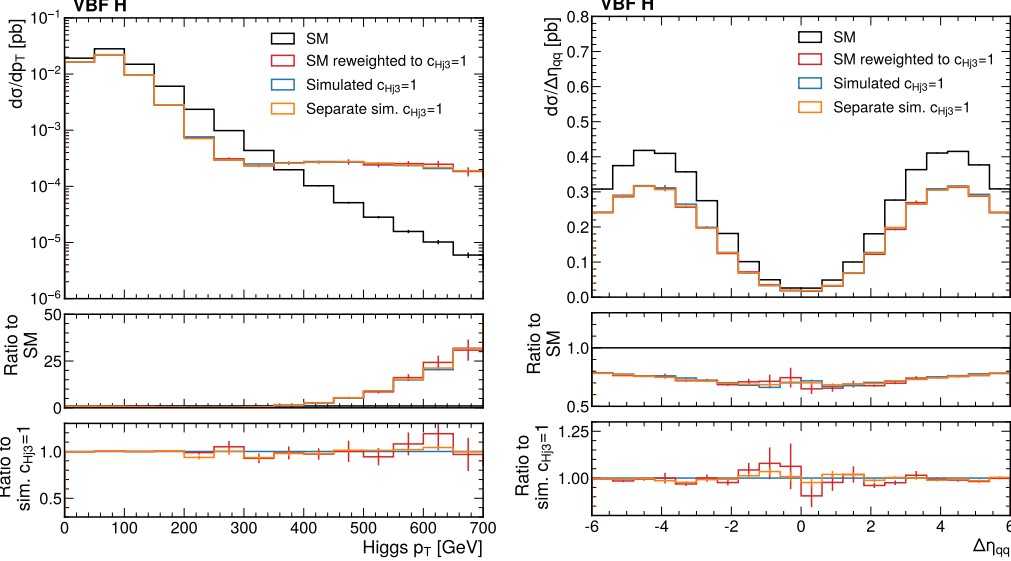

Figure 23: Distribution of (a) the Higgs boson $p_T$ and (b) the angular separation $\Delta\eta$ between the spectator quarks for the $c_{Hj^{(3)}} = 1$ scenario. The SM expectation is shown in black. The prediction for $c_{Hj^{(3)}} = 1$ obtained by reweighting the SM point is shown in red. The direct simulation of $c_{Hj^{(3)}} = 1$ is shown in blue, and the sum of SM, linear-only, and quadratic-only contributions is shown in orange. The middle panel shows the ratio to the SM, and the lower panel shows the ratio to the directly simulated sample.

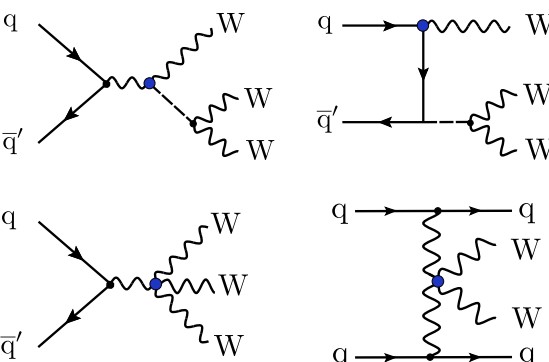

Figure 24: Feynman diagrams for multiboson production with dimension-8 operators (blue markers) affecting the interactions of massive vector bosons, the Higgs boson, and SM quark fields. The bottom left panel shows the production of a pair of W bosons in VBF. Although W boson final states are shown, the diagrams are also valid for Z boson final states.

## 5.7 Multiboson processes

Processes where more than one gauge boson is produced in an LHC collision event or multiboson production constitute a key class of processes at the LHC. These processes can be used to probe the non-abelian gauge structure of the SM and look for effects of new interactions that could modify the SM coupling. ATLAS and CMS measurements cover multiple final states involving pairs or groups of vector bosons. These processes offer a unique window into BSM physics, especially through the EFT framework, allowing direct access to potential anomalous triple and quartic gauge couplings. Multiboson final states are often produced via various quark-initiated parton-level processes. Representative Feynman diagrams are displayed in Fig. 24.

The following choices for the factorization ($\mu_\text{F}$) and renormalization ($\mu_\text{R}$) scales are implemented in MADGRAPH_AMC@LO [51]:

- Option 1: Transverse mass of the $2 \to 2$ system, resulting from $k_\text{T}$ clustering of the final state particles.

- Option 2: Total transverse energy of the event $\sum_{i=1}^{N} \frac{E_i \cdot p_{T,i}}{\sqrt{p_{x,i}^2 + p_{y,i}^2 + p_{z,i}^2}}$, where $N$ denotes the number of decay products.

- Option 3: Sum of the transverse masses $\sum_{i=1}^{N} \sqrt{m_i^2 + p_{T,i}^2}$.

- Option 4: Half of the sum of the transverse masses $\frac{1}{2} \sum_{i=1}^{N} \sqrt{m_i^2 + p_{T,i}^2}$.

- Option 5: Partonic energy $\sqrt{\hat{s}}$.

The default scale choice is typically set to Option 1. However, this choice proves insufficient for the generation of dimension-8 operators, as shown in the left panel of Fig. 25. The dimension-8 operator under consideration is

$$\mathcal{O}_{T,0} = \text{Tr}[\hat{W}_{\mu\nu}\hat{W}^{\mu\nu}] \times \text{Tr}[\hat{W}_{\alpha\beta}\hat{W}^{\alpha\beta}], \tag{20}$$

although the inadequacy of the default scale is independent of the specific operator chosen.

Each process in Fig. 25 is generated separately, representing the SM-only (red-filled histogram), the interference between the SM and BSM components (green-filled histogram), and the BSM-only contribution (blue-filled histogram). The exact syntax used is

```
- generate p p > w+ w+ w- T0=1 (for full generation, shown in black)
- generate p p > w+ w+ w- (SM generation shown in red)
- generate p p > w+ w+ w- T0^2==1
  (Interference between SM and BSM generation shown in green)
- generate p p > w+ w+ w- T0^2==2 (BSM generation shown in blue)
```

where the charge conjugate process was not generated to reduce computation time.

This syntax is also used for both histograms in the left and right panels of Fig. 25. The only difference is the choice of the dynamical scale. The total transverse energy of the event (Option 2) is used for the plot in the right panel of Fig. 25. The total transverse energy provides a better scale choice for generating processes that include dimension-8 operators in triboson production in the bulk of the distribution (right panel), while the tail of the distribution is better predicted by the default scale choice (left panel). One possible explanation for this behavior is that a single scale choice is not appropriate for processes spanning a wide kinematic range. Therefore, the *a priori* assumption that the same scale choice will suffice for both the SM and BSM processes is flawed.

A similar effect is observed for vector boson scattering (VBS) topologies, as shown in Fig. 26. However, in this case, the distribution for the SM process is obtained by reweighting a BSM distribution down to the SM scenario. Other factors that could non-negligibly impact the process generation, such as the PDF choice, are also considered. In the ratio panel, the net effect of the variation of these parameters is shown, highlighting the need to include these effects in analyses as potential sources of systematic uncertainty.

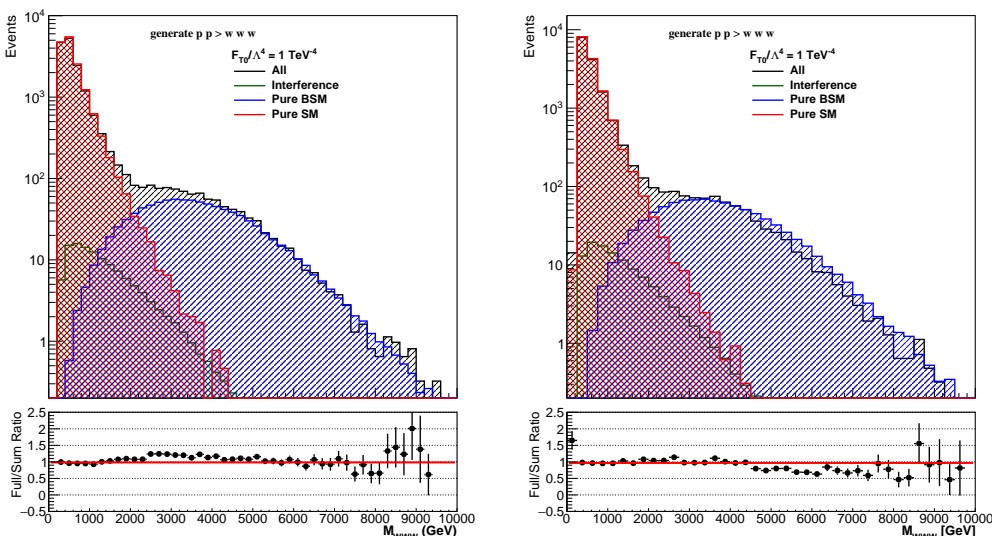

Figure 25: Impact of scale choice in a triboson process. The SM, BSM, and interference terms are generated separately and represented by red, blue, and green hatch-filled histograms, respectively. The full process, generated with all components included, is shown with a black hatch-filled histogram. The syntax of the full process is analogous to the process definition when the reweighting feature of MAD-GRAPH_AMC@LO is used.

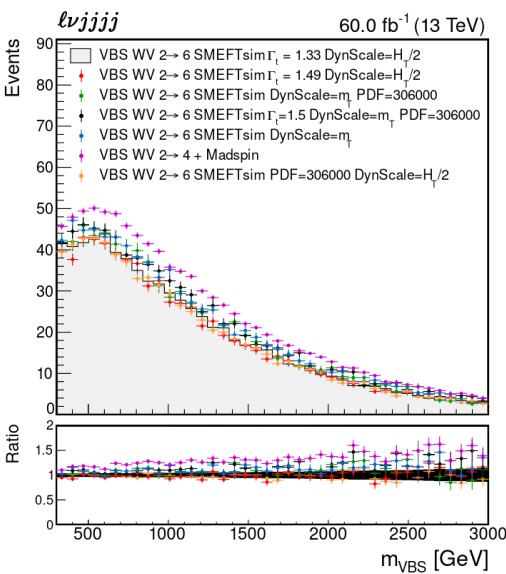

Figure 26: Impact of scale choice in a VBS process. The gray-filled histogram represents the direct generation of the SM term as a 2 → 6 process: p p > e+ ve j j j j QCD=0 NP=0 SMHLOOP=0. The red, green, black, cyan, and orange open histograms show the same process for various generator parameters, including variations in the top width ($\Gamma_t$), different dynamical scale choices, and various PDFs. The magenta points show a further comparison with a 2 → 4 process: p p > v v j j QCD=0 NP=0 SMHLOOP=0 [52].

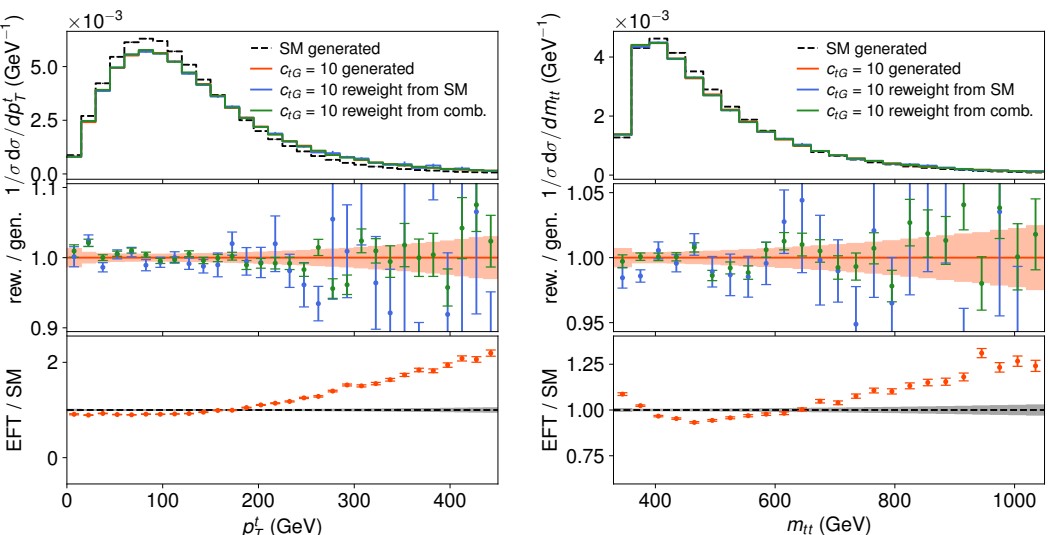

Figure 27: Top quark $p_T$ (left) and $m_{t\bar{t}}$ (right) for SM (black) and SMEFT with $c_{tG} = 10$ for different production methods: Directly generated (orange), reweighted from an SM sample (blue), and reweighted from a combination of SM and other EFT samples (green). Both reweightings were performed post-generation. The panels show the normalized differential cross section (top), the ratio of two reweighting schemes to the direct generation (center), and the ratio of EFT to SM (bottom).

### 5.8 Post-generation reweighting

The reweighting of events is typically done by the same application that generates them, usually immediately after the generation, as part of a single execution. This method was used for all of the previous studies in this report. It requires that the EFT model and the desired WC values (EFT points) are defined at the time of sample generation. However, the need for a new model or different EFT points might arise after the generation, and re-generating a sample can incur significant computing costs, especially when full detector simulation is involved.

Though less common, it is possible to reweight events post-generation, thereby avoiding the significant computing costs of regeneration. This can be achieved by generating an external ME library using MADGRAPH5_AMC@LO. These libraries, provided as Python modules, are specific to a particular model and set of reweighting points. With the LHE-level information from the original generation, a new weight for each reweighting point can be computed by the module.

An advantage of the post-generation approach is that any UFO model can be used as long as the initial and final states match those of the original generation, and the phase space is sufficiently covered. This allows existing SM samples to be reused for EFT analyses, enabling a quick reinterpretation of SM measurements. Similarly, an existing EFT analysis can be easily reinterpreted with new assumptions, such as different flavor structures. Moreover, this method is not limited to MADGRAPH5_AMC@(N)LO-generated samples. As long as the LHE information is available, the reweighting module can be applied to events from any generator.

An example of post-generation reweighting is shown in Fig. 27 for $t\bar{t}$ production using the $c_{tG}$ WC from the dim6top model. A SM $t\bar{t}$ sample is reweighted post-generation to an EFT point with $c_{tG} = 10$ (blue) and compared to a direct generation of the same point (orange).

Furthermore, it is possible to combine multiple reference samples, each reweighted independently to the same EFT point, into a single large sample with higher statistics. This approach is especially useful if the reference samples cover different regions of phase space (though care must be taken to remove events with very large weights). The green line in Fig. 27 illustrates an example where the reweighted prediction is obtained by combining the SM $t\bar{t}$ sample with samples where $c_{tG} = 1$ and $c_{tG} = 3$. This choice improves the population in the tails of the distributions, reducing statistical uncertainty. As seen in Fig. 27, the combined sample provides improved statistics compared to the reweighted SM sample, particularly in the high-energy tails of the distributions, which are relevant for EFT studies.

## 6 Summary

This note serves as a comprehensive guide to simulation strategies within the framework of the Standard Model Effective Field Theory (SMEFT), focusing on the consistency and computational efficiency of various methods. Rather than establishing rigid guidelines, the document evaluates direct event generation and reweighting techniques, offering practical insights for using event generators to study SMEFT effects in complex processes.

Key aspects discussed include the statistical interpretation of simulation- and weight-based strategies, with detailed attention to the balance between accuracy and computational costs. The note compares direct simulations at fixed EFT points, separate matrix element simulations, and reweighting methods, highlighting potential pitfalls like phase space misrepresentation when using nominal samples. It also emphasizes the importance of statistical power in regions where SMEFT operators significantly modify kinematics, especially at high energies.

The intricacies of helicity-aware versus helicity-ignorant reweighting are explored, particularly in WZ, ZH, and $t\bar{t}Z$ production. Helicity-aware reweighting captures subtle SMEFT effects in helicity configurations, while helicity-ignorant reweighting is advantageous in scenarios where SM suppression affects certain helicity states. Both methods are compared for their effectiveness in ensuring accurate predictions, with helicity-aware reweighting proving critical for processes sensitive to angular and polarization effects.

Case studies, such as $t\bar{t}$ and $t\bar{t}Z$ production, demonstrate that reweighting methods generally achieve good closure compared to direct simulations, though phase space coverage limitations can affect predictions in some regions. Additional studies on VBF Higgs production and triboson processes underscore the importance of scale choices in generating EFT samples, with potential impacts on systematic uncertainties. The influence of dimension-6 and dimension-8 operators on kinematic distributions is analyzed, showing good agreement across different simulation methods.

The note provides practical recommendations for improving statistical precision by combining multiple reference samples and reweighting across different EFT points, offering a flexible approach to handling EFT analyses without requiring expensive regeneration. Potential challenges such as large event weights and phase space undercoverage are also discussed in detail, with practical solutions proposed for mitigating these issues.

We hope this note offers a thorough assessment of simulation strategies for SMEFT predictions, emphasizing the importance of selecting the most appropriate method for every use-case. By covering a range of processes, we also hope to provide a solid foundation for future EFT measurements at the LHC, enabling informed choices and ensuring that simulations remain accurate and computationally efficient.

## Acknowledgments

This work was done on behalf of the LHC EFT WG and we would like to thank members of the LHC EFT WG for stimulating discussions that led to this document.

**Funding information**    The work of M. Presilla is supported by the Alexander von Humboldt-Stiftung.

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
