# Peer review of "LHC EFT WG Note: SMEFT predictions, event reweighting, and simulation"

_SciPost Physics Community Reports, doi:SciPost Phys. Comm. Rep. 4 (2024)_

## Round 3 · Referee Report · Anonymous (Referee 1) · 2024-8-13

Report

Dear authors,

many thanks for the well-researched and instructive paper! It was an interesting read and I learnt a lot from it. Overall, it is well written and the content is very much appropriate for this journal. However, before publishing it I would like to ask for a few clarifications, a bit more context at times, and improvements regarding formal aspects. These requests are summarised below. I hope I am not asking for unreasonably time-consuming additions, as I would be happy to see this paper in the journal very soon.

Comments on the content: Fig 3: It is said that the helicity-ignorant reweighting is a bit off at high pT because "statistics is small" (more proper to refer to it as "small sample sizes"), but the error bars in the plot suggest that this is a significant effect. Does this comment refer to statistical fluctuations in the weight calculation? If so, would it be possible to quantify the size of the corresponding uncertainty? This information could e.g. be added as an additional set of error bars in the plot, or by giving the value of the uncertainty in the final bins in the text. If the source of the discrepancy is something else, of course that would be very good to clarify (and quantify). Sec 5.2: - Is there a reason why you mention in the text that pT(H) is an important probe to study BSM effects, but then use pT(Z) in the plots? Of course at LO these may be very strongly correlated, but is there a benefit to showing pT(Z) over pT(H)? (I do not suggest to change any plots, just asking for clarification.) - The description of helicities of reflected Z-bosons and angular distributions is not easy to follow. Would it be possible to e.g. add the plane of reflection into the sketch or show a distribution where the mirrored contribution is apparent? Sec 5.3: - Unlike the previous sections, no PDF set is given here. For completeness, can it be added? - Judging from the bottom left plot of Fig 8, the dedicated points used for ctGRe are not only -0.4, -0.2, 0.2, 0.4 as stated, but also -0.7 and 0.7. Is this correct? - It is not clear to me whether helicity-aware or -ignorant reweighting is used in this section. As this is a focus of the paper, could it be mentioned? (No need to go into comparisons, pros and cons again) - Fig 9: - Would a ratio panel not be useful here? Was it omitted in order to be able to fit the figure into one page? If so, splitting the figure across two pages could be an option. - Having the SM distribution in addition to the sample1 and sample2 may be useful. It can serve as a validation for the “Reweighted to SM” distributions (as in principle, starting from either sample may be off). Moreover, it could help motivate the choice of a BSM starting point for the reweighting by showing that using the SM sample does not give as good agreement/statistical power when reweighting to Pt2. As an SM sample is already available, is this something that can be added? Sec 5.4: - The 5-flavour scheme is mentioned only here and was not mentioned earlier, e.g. in Sec 5.3. Is this worth mentioning? Also, a comment on whether 5FS vs 4FS has any effect on the expected accuracy of EFT reweighting (I assume there is none) may be helpful for readers. - It is mentioned that a comparison between SMEFTsim and SMEFT@NLO is made, using the LO implementation of both packages. However, no results are shown or described. I assume things looked consistent, but can this agreement be quantified? By giving a max deviation in obtained inclusive cross-sections in the text, for example, or a similar measure of consistency. - Fig 11: Is it worth mentioning which helicity configuration(s) are suppressed in the SM sample that lead to the disagreement at high pT when reweighting to cTZ = 1? Sec 5.6: - Fig 16 left: Similar comment as on Fig 3, is it understood why both the reweighting and separate simulation don’t agree so well with the direct simulation at very high pT(H)? Given the error bars, this seems to be a real effect. A comment on how sensitive LHC experiments currently are to a potential bias of this size would also be helpful. - Fig 17-20 (and also 16): If the disagreement in the reweighted distributions at very high pT(H) is due to fluctuations in the computation of weights, it would be good to quantify the corresponding uncertainty. But I find it odd that all the plots in this section have the same rather significant trend at highest pT(H). If it was a downward fluctuation in the SM sample, it would be very unfortunately pronounced, since this appears to be at least a 2sigma effect in every pT(H) plot. And from my understanding, the weights for different Wilson coefficients are calculated independently, so it is also hard to believe that all of them would fluctuate downwards by similar amounts. Could you add a brief summary of your investigations on this? Sec 5.7: - Judging from the mathematical expression, it looks like options 1 and 4 are the same. What is the difference? And could the max sum index N be defined in the text? - Is it correct to use T0=1 but T0^2==1, i.e. both "=" and "=="? - The purpose of showing p p > v v j j in Fig 22 is not clear to me. How does it connect to the point being made about scale choices? Sec 5.8: - In the text, the green histogram in Fig 23 is labeled the SM reweighted histogram, while the plot gives this label to the blue histogram. The green one has much smaller uncertainties, so I assume the label within the plot is correct?

Comments on formalities: - Harmonisation between British and American English, e.g. "parametrization" (Section 1) vs "parameterised" (Section 5.3) - arXiv links in references are broken - "not implemented in [the] MG5aMC re-weighting tool" - Table 1 caption: "thee" - Table 2 caption: remove "multiboson" as these operators are all covered in Table 1? - Clarifying the differences in what plots represent in the plot itself would make things easier to parse. For example in Fig 3, the difference in left and right plots is not evident from the plots themselves. - Fig 4: Should the angle \theta have a \hat? In the sketch there is one, but not in the text. Are these not the same quantity? - At the top of p19 it says "p_t" while otherwise it is denoted "p_T" - MadSpin in Sec 5.5 may want a reference? https://arxiv.org/pdf/1212.3460 - 5.6: - As I did not quite understand the term "transverse mass of the 2->2 system" at first (and naively I would call VBF a 2->3 process), I looked into it and found Eq. (3.1) in https://arxiv.org/pdf/1507.00020 which states that all the final state particles are taken into account and the initial state particles are omitted. If my understanding is correct, could it be rephrased? - 5.7: - Starts with subsubsection 5.7.1, but there is no 5.7.2 - Several instances of a Figure being shown before being mentioned in the text, which can throw a reader off. - Fig 22 caption: "choice of scale choice"

Recommendation

Ask for minor revision

  • validity: -
  • significance: -
  • originality: -
  • clarity: -
  • formatting: -
  • grammar: -

Author:  Robert Schoefbeck  on 2024-10-19  [id 4878]

(in reply to Report 1 on 2024-08-13)
Category:
answer to question

Dear Referee,

We would like to thank you for reviewing this paper and furnishing this report.

We have carefully considered the comments, and we have applied the corresponding changes to the original version of the paper to address the issues raised. A detailed response to the comments can be found in the attached file.

The corresponding document will be arXiv:2406:14620 v4, due on Tuesday, Oct. 22nd.
A resubmission will be done on this day.
We are at your disposal for any further clarifications and/or additional information.

Sincerely,

Robert Schöfbeck, Matteo Presilla, Charlotte Knight, Saptaparna Bhattacharya

Attachment:

LHC_EFT_WG_note_referee_report_1.pdf

---

## Round 3 · Referee Report · Anonymous (Referee 2) · 2024-9-20

Strengths

This report discusses various aspects of SMEFT simulations, providing a useful reference for the community.

In particular the reweighting of samples to new parameter points is discussed in detail.

Several examples from different sectors are discussed.

Weaknesses

Some points could be discussed in a clearer manner (see report).

Some stylistic changes can improve the quality of the report.

Report

The discussion on NLO SMEFT on page 7 is somewhat confusing. The authors talk about both QCD but also QED corrections, it is not clear to me which of the two is causing the problem. Maybe revising that paragraph can help.

Is the "LO+1 jet" sample employed in Section 5.3 a merged sample? Or simply a tt+jet sample? This should be clarified.

Do the authors have uncertainties for Table 3? That would be useful to establish how well the reweighting performs.

I am confused by the discussion of the two different scales in 5.7.1. It is not clear to me how a different choice of scale affects the fact that total=SM+Interference+Squared. Is this what the authors see? Can they clarify this?

Requested changes

Fig. 3 the bottom row: it is worth adding a label within the plots to show what this it for the same sign transverse polarisation.

Tables 1 and 2 appear long before the text refers to them.

Linear and quadratic EFT contributions at NLO can be separated in MG v3. The authors should clarify this.

Could the authors consistently show Feynman diagrams for all processes they consider?

Can the authors make fig. 22 bigger?

Recommendation

Ask for minor revision

  • validity: high
  • significance: good
  • originality: ok
  • clarity: good
  • formatting: good
  • grammar: excellent

Author:  Robert Schoefbeck  on 2024-10-19  [id 4877]

(in reply to Report 2 on 2024-09-20)
Category:
answer to question

Dear Referee,

We would like to thank you for reviewing this paper and furnishing this report.

We have carefully considered the comments, and we have applied the corresponding changes to the original version of the paper to address the issues raised. A detailed response to the comments can be found in the attached file.

The corresponding document will be arXiv:2406:14620 v4, due on Tuesday, Oct. 22nd.
A resubmission will be done on this day.
We are at your disposal for any further clarifications and/or additional information.

Sincerely,

Robert Schöfbeck, Matteo Presilla, Charlotte Knight, Saptaparna Bhattacharya

Attachment:

LHC_EFT_prediction_note_referee_report_2.pdf

---

## Round 4 · Author Response

We thank the referees for the thoughtful comments, which we have addressed and documented in the files attached to each comment's reply.

---

## Round 4 · List of Changes

*) WZ: Fixed a bug effecting events with large EFT weights, improving the consistency of the predictions
*) VBF-H: Fixed scale choice, improving the consistency of the predictions
*) Added representative Feynman diagrams for all processes
*) Improved the clarity of the text in many places
*) VBF-H: Fixed scale choice, improving the consistency of the predictions
*) Added representative Feynman diagrams for all processes
*) Improved the clarity of the text in many places

---

## Editorial Decision

published